# The male bias of a generically-intended masculine pronoun: Evidence from eye-tracking and sentence evaluation

**Theresa Redl** [1,2]*, **Stefan L. Frank**[2], **Peter de Swart**[2], **Helen de Hoop**[2]

**1** Max Planck Institute for Psycholinguistics, Nijmegen, The Netherlands, **2** Centre for Language Studies, Radboud University, Nijmegen, The Netherlands

* Theresa.redl@ru.nl

## Abstract

Two experiments tested whether the Dutch possessive pronoun *zijn* 'his' gives rise to a gender inference and thus causes a male bias when used generically in sentences such as *Everyone was putting on his shoes*. Experiment 1 (*N* = 120, 48 male) was a conceptual replication of a previous eye-tracking study that had not found evidence of a male bias. The results of the current eye-tracking experiment showed the generically-intended masculine pronoun to trigger a gender inference and cause a male bias, but for male participants and in stereotypically neutral stereotype contexts only. No evidence for a male bias was thus found in stereotypically female and male context nor for female participants altogether. Experiment 2 (*N* = 80, 40 male) used the same stimuli as Experiment 1, but employed the sentence evaluation paradigm. No evidence of a male bias was found in Experiment 2. Taken together, the results suggest that the generically-intended masculine pronoun *zijn* 'his' can cause a male bias for male participants even when the referents are previously introduced by inclusive and grammatically gender-unmarked *iedereen* 'everyone'. This male bias surfaces with eye-tracking, which taps directly into early language processing, but not in offline sentence evaluations. Furthermore, the results suggest that the intended generic reading of the masculine possessive pronoun *zijn* 'his' is more readily available for women than for men.

## Introduction

Words with masculine grammatical gender enjoy a special status in many languages. They can be used to refer to men, but they are also often used in a generically-intended way when referring to people whose gender is unknown, unspecified, or when referring to groups of mixed gender. Consider the following headline about the cost and merits of higher education taken from the Dutch tabloid *De Telegraaf*:

1. *Wat kost een student? En wat levert hij op?*

   'How much does a student (MASC.) cost? And how much does he generate?' [1]

**Data Availability Statement:** The data can be found at https://osf.io/cmeub/. The DOI is 10.17605/OSF.IO/CMEUB.

**Funding:** The authors received no specific funding for this work.

**Competing interests:** The authors have declared that no competing interests exist.

We see two instances of such generically-intended masculine words in the example above: the masculine role noun *student* 'student' and the pronoun *hij* 'he'. When reading the newspaper article, it becomes clear that the headline is intended to refer to students regardless of their gender. A question that has occupied researchers for decades is whether such generically-intended masculine word forms trigger a gender inference and make language users think of the referents as male despite the generic intention. Such a male bias has been consistently found for grammatically gender-marked role nouns in languages such as German, French and Norwegian [2–5]. This has been shown using various methods such as sentence evaluation [2], self-paced reading [6], eye-tracking [3] and EEG [5]. All these studies made use of a variation of the following design: A group of people is introduced by means of a role noun in the generically-intended masculine form, and then (a subpart of) the group is revealed to be male or female (e.g., *Die Studenten gingen zur Mensa, weil manche der Frauen/Männer Hunger hatten.* 'The students (MASC.) went to the canteen, because some of the women/men were hungry' [5]). Usually, reading times or ERPs on the anaphor are compared (in this case *Frauen* 'women' versus *Männer* 'men'), with a longer reading time or larger deflection in the ERP component on the female continuation compared to the male continuation taken as evidence for a male bias. In the case of sentence evaluation experiments, participants are asked to indicate whether the second part of the sentence is a good or sensible continuation to the first part, with a higher number of no-responses and longer response times to female continuations—even when they are deemed good continuations—seen as an indication for a male bias. Compared to self-paced reading, eye-tracking and ERP studies, sentence evaluation requires conscious reasoning on the participants' part and therefore taps less into online (and thus the earlier stages of) processing. Nonetheless, taken together, these studies provide overwhelming evidence that generically-intended masculine role nouns indeed lead to an immediate male bias in online processing, at least for the languages that have been tested so far.

For pronouns, the available literature is of a different nature. As criticism of the use of English generically-intended masculine pronouns such as *he* increased starting in the 1970s, the pronoun was put to the test and overwhelmingly found to result in a male bias despite being generically-intended [7–11], with few exceptions [12]. However, these studies used offline methods almost exclusively and therefore did not tap into online processing but only revealed later processes. For example, some researchers asked participants to write a story about a person based on a prompt featuring a generically-intended masculine pronoun [7, 10, 11] while others asked participants to describe their mental imagery after reading or listening to prompts [8, 9]. The more recent research trend of testing generically-intended masculine role nouns with online methods tapping into earlier stages of processing, such as eye-tracking and EEG, has not been extended to the previously heavily researched English pronouns. This can partly be explained by the fact that the use of generically-intended masculine pronouns in English has declined over the years due to intentional changes, and so has research into this phenomenon [13, 14]. However, in other languages, such as Dutch, generically-intended masculine pronouns are still very commonly used and little is known about how they are processed.

Dutch is a particularly interesting language regarding (grammatical) gender and generically-intended masculine words. Similar to some Scandinavian languages such as Swedish, the original Indo-European three-way distinction between feminine, masculine and neuter gender has largely developed into a two-way system distinguishing between common gender (comprising formerly masculine and feminine nouns) and neuter gender. However, the former three-way gender distinction is still visible in 3^(rd) person singular personal as well as possessive pronouns [see also 15]. Furthermore, even though role nouns formally carry common gender, an unambiguously feminine form can often be derived by attaching a feminine suffix such as

-ster (e.g., *schrijver* 'writer'→ *schrijfster* 'female writer'), -es (e.g., *leraar* 'teacher'→ *lerares* 'female teacher') or -e (e.g., *student* 'student'→ *studente* 'female student'). Note, however, that this is not possible for all role nouns (e.g., *arts* 'doctor'), and even if the derivation of a feminine form is morphologically possible, the frequency of feminine word forms varies extensively across role nouns. For example, while *bewoonster* 'female resident' is highly frequent and occurs almost half as often as the masculine counterpart *bewoner* as indicated by two Dutch corpora [16, 17], the frequency of the role noun *kijkster* 'female spectator, viewer' is approximately 1/1000 of that of the masculine equivalent *kijker*. Research shows that Dutch feminine role nouns are gradually becoming less frequent overall [18–20]; consequently, common gender role nouns are increasingly used generically, even though research suggests that they still carry a male bias [21, 22], possibly due to the contrast with the still existing feminine role nouns. Conscious efforts to make Dutch more gender-inclusive regarding role nouns have been taken by several institutions such as the Dutch railway company by eliminating feminine forms [23]. Interestingly, the issue of generically-intended masculine pronouns has received less attention than role nouns in Dutch (as well as in other languages), even though generically-intended masculine pronouns are incompatible with the idea of rendering language gender-neutral. But not only their predominant use and lack of popular alternatives such as English singular *they* [24, 25] or Swedish *hen* [26, 27] make Dutch masculine pronouns an interesting case to study. Dutch is a language with a combination of a grammatical gender and natural gender system [28]. Audring [15] explains that this development from a grammatical to a natural gender system (or in her words: from a syntactic to a semantic gender agreement system) gave Dutch masculine pronouns an interesting default status beyond reference to people. For example, inanimate objects such as a bicycle, which has common gender in Dutch (*de fiets*), can also be referred to with a masculine pronoun, e.g. *Hij staat in de schuur* (lit. 'He is standing in the shed'). This increased default status of Dutch masculine pronouns as well as their ubiquity in daily language use for generic reference to humans make them a particularly interesting case to study.

In an effort to answer the question whether generically-intended masculine pronouns lead to an online and immediate male bias in processing in Dutch, Redl, Eerland, and Sanders [29] conducted an eye-tracking experiment targeting the possessive pronoun *zijn* 'his'. With their design, they followed studies on generically-intended masculine role nouns, the majority of which simultaneously tested the effect of gender stereotype information and the effect of the role nouns' masculine gender on the mental representation of the referent's gender [e.g., 2–4, 6, 30, 31]. Similar to the grammatical gender of generically-intended masculine words, the gender stereotypicality of role nouns (e.g., *a carpenter is male*) has been shown to be readily used to infer a referent's gender. For example, Duffy and Keir [32] presented native speakers of English with sentences such as *The babysitter found himself humming while walking up to the door*. Reading times on the reflexive pronoun and the subsequent region were significantly higher when the pronoun gender did not match the stereotype of the role noun, as in the example above. Thus, the stereotype information triggered a gender inference when it was encountered, which in turn led to processing difficulties when the pronoun revealed the referent's gender to mismatch the stereotype. In a second experiment, these target sentences were preceded by a context consisting of multiple sentences already introducing the referent using the role noun as well as explicitly stating whether the referent (i.e., the babysitter in the example above) was a man or a woman. When the target sentence was presented after such a context, the mismatch effect was eliminated. Similarly, Carreiras, Garnham, Oakhill et al. [33] tested sentence pairs such as *The babysitter settled down to watch a video. Then he heard the baby crying*, in both English and Spanish. The results of their self-paced reading experiment on English showed that reading times for the second sentence increased when the pronoun did

not match the role noun's gender stereotype, as in the example above. Thus, again, readers used the gender stereotype information to make a gender inference and subsequently slowed down when the person's gender was revealed by the pronoun and mismatched that stereotype. The results of the Spanish experiments provided additional insights. In Spanish, the determiner *el/la* preceding the role noun already indicates whether the referent is male or female. The results showed that Spanish readers showed higher reading times on the *first* sentence when there was a gender mismatch, but reading times on the *second* sentence featuring the pronoun were similar across conditions. This shows that stereotype information is automatically processed as soon as it becomes available, even in the presence of more reliable gender cues. However, once this mismatch is resolved, pronouns mismatching the stereotype (but matching the previously established referent's gender) do not lead to processing difficulties anymore [for further evidence of the automatic activation of gender stereotypes from English see 34–36].

As mentioned above, studies on generically-intended masculine role nouns often used stereotypically female, male and neutral role nouns to investigate the effect of gender stereotypes and the male bias of the grammatical gender at the same time [e.g., 2–4, 6, 30, 31]. Researchers can then test whether generically-intended masculine words cause a male bias in the absence of other gender information, as well as in stereotypical contexts that can trigger a gender inference in their own right. Redl et al. [29], too, embedded their target pronoun *zijn* 'his' in stereotypical contexts, which were provided by stereotypical activities instead of role nouns. They constructed sentences featuring the possessive pronoun *zijn* 'his' in generically-intended contexts, and analyzed early as well as late measures of reading on a male or female proper noun later on, for example:

2. *Iedereen was **zijn** veters aan het strikken. Zo was ook **Maaike/Stefan** zich aan het klaarmaken om naar buiten te gaan.*

'Everyone was tying his shoelaces. Maaike (f)/Stefan (m) was also getting ready to go outside.'

They varied the activity in the first sentence, which was expressed by a verb and a direct object, between stereotypically neutral contexts (e.g., *tying shoelaces*), as well as stereotypically female (e.g., *doing yoga exercises*) and male contexts (e.g., *practicing soccer tricks*). This was done to test whether a male bias surfaced at all, which would be most clearly seen in stereotypically neutral contexts (i.e., in the absence of biasing gender stereotype information), and whether this male bias persisted in contexts that provided additional gender information in the form of stereotypes—a type of gender cue that has been repeatedly shown to trigger gender inferences. Contrary to their hypotheses, they found no evidence for a male bias of *zijn* 'his'. Put differently, they did not find an increase for any of the analyzed reading time measures on female proper nouns in any of the contexts. They did, however, find that the violation of a gender stereotype led to an increase in dwell time (also called total fixation duration) on the proper noun, but for male protagonists transgressing expectations based on stereotypes only. Redl et al. [29] offered various possible explanations for why no evidence of a male bias of *zijn* 'his' was found. First, it could be that the generically-intended masculine pronoun *zijn* 'his' simply does not trigger a gender inference. It would then function as intended, namely as a true generic, since it does not make it more likely for referents to be mentally represented as male than female. As explained above, the masculine grammatical gender enjoys a wide-reaching default status in Dutch, which might make the generic reading more readily available in online processing. The generic potential of *zijn* 'his' might have been boosted by the fact that the referents were first introduced by gender-unmarked and inclusive *iedereen* 'everyone',

after which the pronoun *zijn* 'his' was introduced and anaphorically linked to *iedereen* 'everyone'. A second possible explanation is that the generically-intended masculine pronoun leads to a male bias, but it was not detected in the eye-tracking experiment. According to Redl et al., one possibility for why the hypothesized male bias might not have surfaced lies in the method. The participants' only task during an eye-tracking experiment such as Redl et al.'s is reading; processing difficulties can be inferred from looking patterns and reading times, and the method therefore directly taps into automatic language processing. As explained above, studies on English *he* and *his* have found that these pronouns do lead to a male bias, but they have done so using methods that required more conscious processing and evaluation of what is being read than in the eye-tracking experiment by Redl et al., thereby tapping into later processing stages. In a similar vein, using the sentence evaluation paradigm, Garnham, Gabriel, Sarrasin, Gygax and Oakhill [31] and Gygax et al. [2] found that the grammatical gender of role nouns overrode stereotype information in German. Thus, they found generically-intended masculine role nouns to induce a male bias across neutral, female and male stereotype contexts when participants had to evaluate these sentences. Stereotypes showed no additional effect. Studies on German using the self-paced reading and eye-tracking methods, however, found gender stereotype information to have a moderating effect [3, 6]. It is thus possible that a male bias is more likely to surface when employing a task which requires more conscious processing and the evaluation of a stimulus (e.g., sentence evaluation) than a task tapping into automatic language processing (e.g., reading during eye-tracking).

Given the large number of studies which found English masculine pronouns as well as masculine role nouns across languages to cause a male bias when used generically, we want to follow up on the null result by Redl et al. [29]. Therefore, the primary goal of the experiments presented in this paper is to test the same hypothesis formulated by Redl et al., namely that the possessive pronoun *zijn* 'his' leads to a male bias in language processing. If, however, we were to replicate the null results found by Redl et al., this would add to the evidence suggesting that the masculine gender of *zijn* 'his' does not trigger a gender inference and that the pronoun can in fact function as a true generic, at least when used in combination with the indefinite pronoun *iedereen* 'everyone'. The secondary goal of these experiments is to investigate whether the surfacing of a male bias is facilitated by a method requiring more conscious processing of the pronoun. To this end, we conducted two experiments. First, we conducted a conceptual replication of the eye-tracking experiment by Redl et al. to test for a male bias of *zijn* 'his' after making several changes to their eye-tracking design. Second, we conducted a sentence evaluation experiment similar to Gygax et al. [2] and Garnham et al. [31] to test whether the male bias of *zijn* 'his' surfaces using a method which requires participants to process the generically-intended masculine pronoun more consciously and go beyond mere automatic (and thus the early stages of) processing. The same stimuli were used in both experiments.

We further strove to balance our sample between female and male participants as much as possible and include their gender in the analysis in both experiments. Redl et al. found no evidence of a male bias for either gender. However, they did find a significant interaction effect involving participant gender on the spillover region; female as well as male participants showed a slight increase in dwell time when the mentioned individual shared the participants' gender. While this effect was very weak and did not hint at a male bias caused by the pronoun, it suggests that reading time differences between the genders can arise as a result of the experimental manipulation at hand. Thus, testing the effect of participant gender should not be overlooked in follow-up studies. Furthermore, past research has shown that men are more likely to experience a male bias when confronted with a generically-intended masculine word. In their meta-analysis of 19 studies on English, which predominantly investigated generically-intended masculine pronouns, Henley and Abueg [37] found that almost all studies reported a

difference in male bias by participant gender. They state that "in all these studies finding a significant difference, females were found to interpret generics more generically . . . than did males, on at least one, and usually all, measures" [37]. Their meta-analysis found this effect to be highly robust. Citing Nilsen [38], Henley and Abueg suggest that this effect is possibly rooted in language acquisition. While boys and men are included as referents whenever masculine word forms are used, girls have to learn that language forms mismatching their own gender can still be intended to refer to them. These results on English generically-intended masculine pronouns make it paramount to study the effect of participant gender when investigating this phenomenon in other languages, which we therefore did in the two experiments reported below.

## Experiment 1: Eye-tracking during reading

This eye-tracking experiment was a conceptual replication of Redl et al. [29]. As described above, the goal was to test if evidence of a male bias induced by the possessive pronoun *zijn* 'his' would be found using a highly similar, but slightly adapted design, or if again no male bias induced by *zijn* 'his' would be found when repeating the experiment. The basic scaffolding of the materials remained the same, which meant that the possessive pronoun *zijn* 'his' was embedded in contexts that allowed for a male-specific reading (i.e., referring to men only) as well as for a generic reading (i.e., referring to people of all genders). As we were interested in the effect of the pronoun, we did not include any other gender-marked words in the stimuli in order to avoid possible confounds. This excluded role nouns as antecedents, as they have been shown to lead to a male bias [21, 22], making it hard to disentangle a possible male bias of the pronoun from that of the role noun when used in the same stimulus [21, 22]. Like Redl et al., we therefore opted for the pronoun *iedereen* 'everyone' as an antecedent for the possessive pronoun *zijn* 'his'. Since *zijn* 'his' is often used generically in combination with the indefinite pronoun *iedereen* 'everyone' [see Chapter 1 in 39 for numerous corpus examples], the use of such stimuli allows us test if the intended reading in such frequently occurring contexts is actually achieved. As mentioned above, introducing the referents by means of grammatically gender-unmarked and inclusive *iedereen* 'everyone' and using the pronoun *zijn* 'his' anaphorically, might facilitate a generic interpretation of the masculine pronoun. Thus, if a male bias is found in the contexts we used, then this strongly suggests that the male bias would also surface in other, less inclusive contexts, for example when reversing the order of *iedereen* 'everyone' and *zijn* 'his' (i.e., a cataphoric—as opposed to anaphoric—use of *zijn* 'his').

In summary, the choice of comparing *iedereen* 'everyone' with *zijn* 'his' remained the same compared to the stimuli in the study by Redl et al. [29]. However, we made several other changes to the initial design. First of all, we selected new activities based on a large-scale rating study, which is detailed below. By doing this, we were able to counterbalance the stereotypically female and male activities even more, and also select stereotypically neutral activities with a mean even closer to the middle of the scale. Twenty activities were the same as those from Redl et al., and an additional 52 new ones were used, all 72 based on the new rating study. Furthermore, several changes were made to the continuation of the sentence. A direct comparison of two stimuli featuring the same activity in Redl et al. and the current study can be seen in Table 1.

Instead of using proper nouns to indicate the referents' gender, we used the nouns *vrouwen* 'women' and *mannen* 'men' to unambiguously identify the subgroup. This was done to eliminate variance induced by frequency effects of the proper nouns. Furthermore, the group of people was introduced by the connective *waaronder* 'among whom', thereby more explicitly indicating membership status. A pre-test was conducted to assure that this reading was indeed

**Table 1. Example stimuli for the current eye-tracking experiment and Redl et al. [29] with the most important changes in bold.**

| Study | Example stimulus |
|---|---|
| *Current experiment* | *Iedereen was zijn veters aan het strikken,* **waaronder een paar vrouwen/mannen** *die al tien minuten geleden hadden moeten vertrekken, maar zich hadden verslapen.* |
| | 'Everyone was tying his shoelaces, **among whom a few women/men** who would have had to leave ten minutes ago, but had overslept.' |
| *Redl et al. (2018)* | *Iedereen was zijn veters aan het strikken.* **Zo was ook Maaike/Stefan** *zich aan het klaarmaken om naar buiten te gaan.* |
| | 'Everyone was tying his shoelaces. **Maaike (f)/Stefan (m) was also** getting ready to go outside.' |

achieved. This membership had not been made explicit in the stimuli of Redl et al. It is therefore theoretically possible that the individual in the second sentence was not interpreted as being part of the group referred to by *zijn* 'his' in their experiment. The male bias of the pronoun would then not arise when referring to a female individual in the second sentence.

The sentences used in the current experiment were also longer than those in Redl et al. [29]. This was mainly done to distract from the repetitive pattern and make the sentences more informative and engaging. We also decided to present the stimuli on the screen in a way that allowed us to also analyze reading times on the pre-view region of the noun indicating a gender (mis)match with the pronoun. In an eye-tracking experiment on German, Irmen [3] introduced a group of people by means of a generically-intended masculine role noun (e.g., *Nachbarn* 'neighbors (MASC.)') and then referred back to them by means of the noun phrase *diese Frauen/Männer* 'these women/men'. She treated the whole noun phrase as her main region of interest, but analyzed the determiner and the noun separately. A spillover region was analyzed as well. Irmen found an effect of stereotype on the determiner as well as on the spillover region, while the effect of grammatical gender (i.e., the male bias induced by the generically-intended masculine role noun) was found on the noun itself. This shows that effects of incongruent gender information can already affect reading times on the word preceding the noun identifying (some of) the referents' gender. The decision to include this pre-view region in the analysis was further informed by research into reading patterns, showing that semantic information up to six to eight characters to the right of the current fixation is processed [40, 41].

The original six conditions in Redl et al. [29] resulted from the manipulation of the stereotype context between neutral, female and male and manipulating the continuation between female and male. In the present study, we further included a control condition for each of the six original conditions. The six control conditions were very similar to the experimental conditions, but we replaced the singular masculine pronoun *zijn* 'his' with the plural gender-unmarked pronoun *hun* 'their', for example:

3. *Ze waren allemaal* **hun** *veters aan het strikken, waaronder een paar vrouwen/mannen die al tien minuten geleden hadden moeten vertrekken, maar zich hadden verslapen.*

'They were all tying their shoelaces, among whom a few women/men who would have had to leave ten minutes ago, but had overslept.'

This allowed us to more clearly identify a possible effect of the masculine pronoun, as the only relevant difference between the control and the experimental conditions is the masculine pronoun itself. The combination of *ze* 'they' with *allemaal* 'all, altogether, everybody' as the antecedent of *hun* 'their' was intended to convey a similarly inclusive meaning as *iedereen* 'everyone'. The possessive pronoun *hun* 'their' in the control condition, however, is not

marked for grammatical gender. Thus, the control conditions do not contain any grammatical gender information in the first clause. By comparing the stereotypically neutral control condition to the experimental condition featuring masculine *zijn* 'his', we can make absolutely sure that any found effect is truly due to the pronoun and not due to a more general male bias [3, 42], or due to other lexical differences between the target words. Even if the stereotypically neutral control condition is biased towards men due to a more general male bias [3, 42], we would be able to isolate the male bias of the pronoun by comparing the control to the experimental conditions in the analysis. Furthermore, the addition of control conditions for the stereotypically male and female experimental conditions as well, enables us to tease apart the possible effect of the pronoun from any (added) effect of the stereotypes, shedding further light on how these two different types of gender cues interact.

We further decided to include participant gender as an independent variable in all models as opposed to Redl et al. [29] who only added this factor when it significantly improved the model fit. As explained in the Introduction, many studies on English generically-intended masculine pronouns have found that women and men can differ regarding the presence and strength of the male bias [7, 8, 11, 43]. More specifically, these experiments revealed that men showed a stronger male bias, and that women were more likely to arrive at the generic reading. Like Redl et al., we hypothesized that the generically-intended masculine pronoun *zijn* 'his' would cause a male bias which would be moderated by the gender stereotype context. We further hypothesized that if men and women were to differ in their processing of the generically-intended masculine pronoun, male participants would show a larger bias.

## Materials and method

**Materials.** We conducted a large rating study to identify stereotypically neutral, female and male activities to be used in the stimuli. We conceived of 363 potentially stereotypical activities through intensive brain-storming sessions. All activities had to be expressed by a transitive verb and a direct object (e.g., *tanden poetsen 'brushing teeth'), so that a pairing with the possessive pronoun zijn 'his' was syntactically possible. We further tried to test as many activities as possible which would lead to a distributive reading (i.e., everyone brushing their own teeth) instead of a collective reading (i.e., everyone brushing the teeth of the same male individual). The distributive reading is necessary for the use of* generically-intended *zijn 'his' to be felicitous. While* Redl et al. [29] had run a smaller rating study with 123 activities, we decided to find and test a larger number of potential stereotypical activities to be able to balance our stereotype conditions even more with regard to their strength. The rating study's design is also very similar to that of previous studies which tested the stereotypicality of role nouns such as surgeon [44–47]. As previous studies have found that participant gender as well as scale direction affect stereotype ratings, we controlled for their influence.

A total of 56 native speakers of Dutch (28 male) ranging in age from 18 to 30 ($M = 20.5$) completed the online questionnaire, which was administered through Qualtrics [48]. Participants who were recruited through the Radboud Research Participation System SONA received course credits. A smaller number of participants was recruited through Facebook and did not receive reimbursement. Participants were presented with a list of 363 activities in a fully randomized order. The stereotypicality of these activities had to be rated on a 7-point Likert scale ranging from −3 to +3. Participants were asked to indicate for each activity how likely it was that the activity was carried out by a man or a woman. It was emphasized that they should provide a rating reflecting their perception of reality, and not a rating that reflected their idea of an ideal world. The main part of the questionnaire featuring the activities was divided into two lists, which differed in scale direction. Male and female participants were evenly distributed

across lists, with 14 female and 14 male participants being asked to rate all 363 activities with −3 corresponding to *female*, and the other 14 male and 14 female participants being asked to rate the activities with −3 corresponding to *male*.

The ratings ranging from −3 to +3 were automatically coded as values ranging from 1 to 7 by Qualtrics. Furthermore, all ratings were (re)coded so that *1* corresponded to a fully female interpretation. For each of the 363 activities, the mean across all participants and the standard deviation were calculated. All activities, their translation, their mean ratings and standard deviation can be found in the (S1 Table). The mean ratings for the activities ranged from 1.2 for *meidenavond plannen* 'planning a girls' night out' to 6.8 for *mannenavond plannen* 'planning a guys' night out'.

Based on this rating study, 32 stereotypically neutral, 32 stereotypically female and 32 stereotypically male activities were selected. A total of 96 stimuli were initially designed and subjected to two further pre-tests, based on which we chose 72 stimuli to be used in the experiment. All stimuli can be found in the S1 Stimuli. The two pre-tests ensured that our items were perceived as plausible (Pre-test 1) and that the group of women or men was actually perceived as a subpart of the group mentioned in the beginning of the sentence (Pre-test 2). For more information on the pre-tests, see the (S1 Pre-tests). The final selection of 72 activities for the stimuli can also be found in the (S2 Table). All stereotypically female activities had a mean rating of 3 or lower on a scale from 1 (female) to 7 (male), stereotypically male activities had a mean rating of 5 or higher, and neutral stereotypes had a rating of approximately 4. We chose stereotypically female and male activities such that their range was similar ([1.71; 2.88] for female stereotypes, [1.95; 2.95] for male stereotypes when reversing the scale), their mean was similar (2.26 versus 2.27) and the distribution within that range was similar (i.e., similar standard deviations: 0.33 versus 0.31). The stereotypically neutral activities had a mean of 3.99 and a standard deviation of 0.13, ranging from 3.80 to 4.25. All chosen activities had a standard deviation below 1.

All experimental items followed the same pattern as the example in Table 1 above. Just like [29], we introduced a group engaging in an activity by means of the indefinite pronoun *iedereen* 'everyone'. Each stereotypical activity consisted of a verb and a direct object. The activities were varied between stereotpically neutral (e.g., *schoenen aandoen* 'putting on shoes'), female (e.g., *yogaoefeningen doen* 'doing yoga exercises') and male (e.g., *voetbaltrucs oefenen* 'practicing soccer tricks') and occurred in a progressive form. This first part of the sentence included the generically-intended masculine pronoun *zijn* 'his'. A subpart of this group was then explicitly identified as being female or male (*waaronder een paar vrouwen/mannen* 'among whom a few women/men') and more information about them was provided. We varied *enkele vrouwen/mannen* 'some women/men' with *een paar vrouwen/mannen* 'a few women/men'. The two noun phrases are highly similar in meaning and were not expected to affect the results in any way. We also included control items which featured the gender-neutral plural pronoun *hun* 'their' rather than masculine *zijn* 'his' (see the example in (3) above). The meaning of the two sentence types was maximally similar while the one featured the generically-intended masculine possessive pronoun and the other did not. Every stereotypical activity was thus embedded in eight different stimulus versions, due to the variation of pronoun (*zijn* 'his' versus *hun* 'their'), continuation (*vrouwen* 'women' versus *mannen* 'men') and the variation between the quantifiers *enkele* 'some' and *een paar* 'a few', the latter variation not being experimentally relevant. Consequently, eight lists to which participants were randomly assigned were created making use of a Latin Square design.

Of the 146 fillers, 18 were stimuli for a different experiment [Experiment 3 in 49]. The remaining 128 fillers were designed to resemble the experimental stimuli in terms of complexity, but did not feature *iedereen* 'everyone', *zijn* 'his', *hun* 'their', *vrouw/vrouwen* 'woman/

women' or *man/mannen* 'man/men' and were as neutral as possible regarding stereotypes. Each participant saw experimental and filler items in a different pseudo-randomized order, which was created by means of the program Mix [50].

To check whether participants read the stimuli attentively, statements about stimuli were displayed after 24 of the 72 stimuli and 46 of the 146 fillers. The statements had to be judged as correct or incorrect. Half of the statements were correct and half were incorrect, equally distributed over fillers and experimental items as well as over conditions.

**Participants.** 121 native speakers of Dutch (48 male) were tested. They ranged in age from 18 to 29 (M = 22.1). The majority of participants were students (N = 107). Participants received a coupon worth €10 or course credit when preferred. Two exclusion criteria applied: data of participants who guessed the purpose of the experiment or who responded incorrectly to more than 20% of the comprehension statements were not considered in the analysis. None of the participants correctly guessed the purpose of the experiment. One participant had to be excluded, as she responded incorrectly to more than 20% of the comprehension statements, leaving 120 participants (48 male), aged 18 to 29 (M = 22.2), for the analysis. All participants gave written consent. The research presented in this article was approved by the Ethics Assessment Committee Humanities of the Radboud University (number 4592).

**Apparatus.** The experiment was conducted at the Centre for Language Studies Lab at Radboud University. We used an EyeLink 1000+ remote desktop eye-tracker with a head stabilizer to minimize head movements. Data were recorded with a sampling rate of 1000Hz. The stimuli were presented using the software Experiment Builder by SR Research [51], on a BenQ XL 2420T 24" screen, but the used resolution was set to 1024×768. The distance between participant and screen was 108cm. The stimuli were presented in black letters on a gray background using the font Calibri with a font size of 19.

## Analysis

The raw eye-tracking data were pre-processed using EyeLink Data Viewer by SR Research. The fixation pattern of each item was manually checked for each participant. Given a systematic and clear drift in a trial, the fixations were reassigned in accordance with the drift. In addition, if the first fixation of a trial did not fall on the first line of the stimulus, but the subsequent fixations did, the initial fixation was deleted. Using Data Viewer's clean-up procedure, fixations that were smaller than 80ms were merged with another fixation within 0.25 degrees in visual angle on the x-axis if this fixation exceeded 80ms (this translates to approximately 0.47cm on the screen). In a second step, fixations that were larger than 1200ms or smaller than 80ms and could not be merged were deleted. Then, three reading time measures were calculated for the regions of interest: first run dwell time (i.e., the sum of the duration of all fixations in a region when it is entered for the first time, also known as gaze duration), regression path duration (i.e., first run dwell time with the addition of the duration of fixations back to previous regions out of the analyzed region) and dwell time (i.e., the sum of the duration of all fixations in a region, also known as total fixation duration). We decided to include early as well as late measures of reading, as earlier eye-tracking studies have shown that both grammatical gender and gender stereotypes can affect the reading process from early to late [3, 52]. We opted for the same reading time measures as Redl et al. [29], except that we omitted first fixation duration. This was done since previous studies have either found no reliable effects for first fixation duration [29, 52] or found the results for first fixation duration to be similar to those regarding first run dwell time [3]. First run dwell time is regarded as the upper bound of earliest processing, with regression path duration and dwell time being indicative of later processes [53]. The regions of interest were defined as indicated by the square brackets:

4. *Iedereen was zijn schoenen aan het aandoen, waaronder [enkele/een paar]₁ [vrouwen/man-nen]₂ [die al]₃ bijna klaar waren om de deur uit te gaan, maar een beetje aan het treuzelen waren.*

'Everyone was putting on his shoes, among whom [some/a few]₁ [women/men]₂ [who already]₃ were almost ready to go out, but were still slacking.'

As explained above, we opted to analyze the quantifier preceding the gendered noun *vrouwen/mannen* 'women/men', since semantic information up to six to eight characters to the right of the current fixation is processed [40, 41]. Region 3 functioned as a spillover region. All analyzed regions are in the second clause. We chose to focus on the second clause and not to analyze measures of regression into regions in the first clause, because regression path duration on the three chosen regions by definition—as it is provided above—already capture regressions into the first clause. Exploratory analyses on the number of regressions into the first clause were conducted after the main analysis and can be found in the Supporting information.

The data were analyzed in R version 3.5.2 [54] using the *lmer* function from the *lme4* package [55]. All described models were fitted to log-transformed reading times to correct for a right skew in the data, as determined by visually inspecting histograms of the data before and after transformation. Model residuals were also checked for normality by means of Q-Q plots. As the primary research question of this experiment was whether the possessive pronoun *zijn* 'his' leads to a male bias, we decided to first only analyze the neutral stereotype contexts, as a male bias could be most easily isolated and seen in these contexts if present at all. This choice was made in order to simplify the analysis. Due to the presence of four factors and twelve conditions in total, four-way interactions could possibly arise, but they are notoriously hard to interpret. Note that Redl et al. [29] only had three possible factors and six conditions due to the absence of elaborate control conditions. Our approach is thus different from their statistical analysis. In a first step, PARTICIPANT GENDER (*female* versus *male*), PRONOUN (*zijn* 'his' versus *hun* 'their') and CONTINUATION (*vrouwen* 'women' versus *mannen* 'men') served as fixed effects. Only if a male bias was found (i.e., an interaction between PRONOUN and CONTINUATION, such that the reading time increase for female compared to male continuations is larger on experimental than on control items) for either or both participant genders (i.e., for all participants, or only for male participants or—though we hypothesized this to be less likely—only for female participants), did we extend the analysis to stereotypically male and female contexts as well. This was done to investigate if the male bias of the pronoun persists in otherwise gendered contexts. If no male bias was found in the stereotypically neutral context, we did not extend the analysis to female and male stereotype contexts.

The factors were coded using simple contrasts. Simple contrasts are similar to dummy or treatment contrasts in the sense that a reference level can be chosen. However, the intercept represents the mean of means. The reference level is coded as $-1/k$, with $k$ being the number of levels of a factor. The level to be contrasted with the reference level is coded as $(k-1)/k$. This means that the simple contrast coding for two-level factors is the same as for sum or deviation contrasts. For CONTINUATION, *vrouwen* 'women' was coded as ½, *mannen* 'men' was coded as $-$½. For PARTICIPANT GENDER, the female participants were coded as ½, male participants as $-$½. For PRONOUN, *hun* 'their' was coded as ½, *zijn* 'his' was coded as $-$½. With STEREOTYPE being a three-level factor, two different contrasts were defined, one comparing the female and the neutral level (female = ⅔, male = $-$⅓, neutral = $-$⅓), the other comparing the male and the neutral level (female = $-$⅓, male = ⅔, neutral = $-$⅓).

Initially, the full random structure permitted by the design was fitted. In cases of non-convergence, we first suppressed the correlation parameters. We then checked for overparameterization

using Principal Component Analysis from the *RePsychLing* package and removed components which explained little to no variation and were therefore negligible, starting with higher order terms in accordance with Bates, Kliegl, Vasishth, and Baayen [56]. All models included random intercepts for participants and items. The random slope structure of the final models is reported below. *P*-values were obtained using the *lmerTest* package [57].

We followed Benjamini and Hochberg [58] and applied their false discovery rate (FDR) control in order to correct for conducting multiple tests, as we analyzed three reading time measures in three regions, leading to nine models or tests. The inflated Type I error rate caused by multiple comparisons is traditionally ignored in eye-tracking research, but Von der Malsburg and Angele showed [59] with Monte Carlo simulations that "false positives are increased to unacceptable levels when no corrections are applied" (p.133). We therefore applied the correction by Benjamini and Hochberg [58] in order to reduce the chance of a false positive. Only *p*-values smaller than the FDR-corrected threshold are reported below. The full models can be found in the (S1 Model summaries Experiment 1).

## Results

**Region 1: Quantifier.** We found a significant effect for first run dwell time. The final model included random slopes for PRONOUN for participants, and PARTICIPANT GENDER and the interaction between PARTICIPANT GENDER and CONTINUATION for items. We found a significant three-way interaction between CONTINUATION, PRONOUN and PARTICIPANT GENDER ($\beta = 0.16$, SE = 0.05, t = 3.07, p = 0.002). As can be seen in Fig 1, male participants showed a significant increase for the continuation women, but only in the zijn 'his' condition. As no other explicitly biasing gender information is provided in stereotypically neutral contexts, this can be seen as an indication that zijn 'his', though intended as generic, does indeed lead to a male bias. This is further supported by the fact that this difference between women and men as continuations was not found in conditions with the genderless pronoun *hun 'their'. No significant differences were found for female participants.*

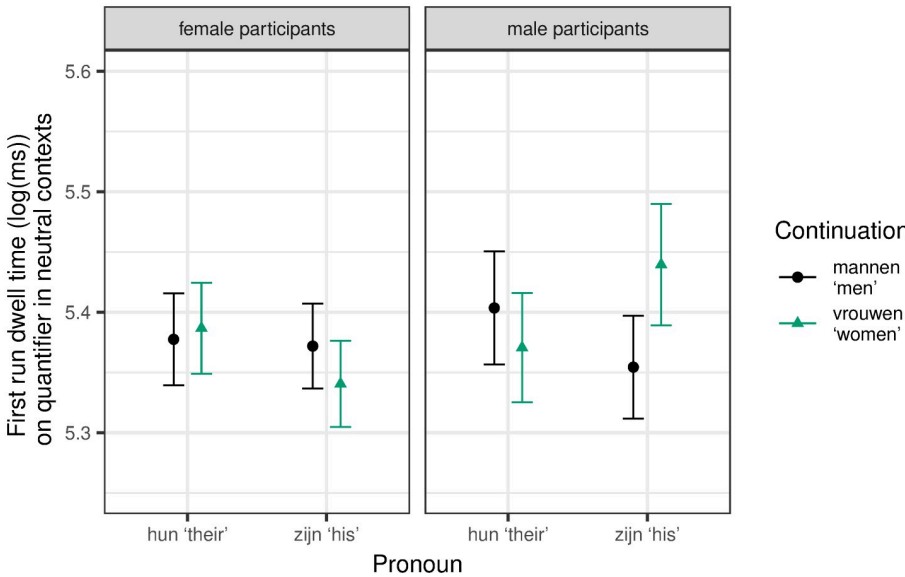

**Fig 1. Mean log-transformed first run dwell time on Region 1 (quantifier) with 95% within-subject confidence intervals based on Morey [60].**

We extended the analysis to female and male stereotype contexts to see if the effect of PRO-NOUN persists in otherwise gendered contexts, but for male participants only. See Fig 2 for the results. The final model included random slopes for CONTINUATION, PRONOUN, PRONOUN*STEREO-TYPE (*male* versus *neutral*) and PRONOUN*CONTINUATION*STEREOTYPE (*male* versus *neutral*) for participants, and random slopes for PRONOUN for items. *P*-values were controlled for multiple tests by subjecting the *p*-values of the three effects of interest (i.e., PRONOUN*CONTINUATION and PRO-NOUN*CONTINUATION*STEREOTYPE (both contrasts)) to Benjamini and Hochberg's [58] procedure. There were no significant effects after applying this correction. Notably, we should see higher reading times for the word *women* across all stereotype contexts in the *zijn* 'his' conditions if the male bias had persisted throughout all stereotype contexts. This is not the case. If the male bias persisted, but the stereotype context had an effect, too, we would see an even bigger difference between the continuations *women* and *men* in male stereotype contexts (due to a "double" male bias), but a smaller difference in female stereotype contexts. This is not the case either. It rather seems as if *zijn* 'his' does not lead to a male bias in otherwise gendered contexts.

No significant results were found for regression path duration and dwell time.

**Region 2: Noun.**   No significant effects were found for any of the reading times on the second region *vrouwen/mannen* 'women/men'.

**Region 3: Spillover.**   No significant effects were found for any of the reading times on the spillover region.

To summarize the results, we found a significant three-way interaction between CONTINUA-TION, PRONOUN and PARTICIPANT GENDER on the quantifier for first run dwell time. This suggests that men, but not women, experienced a male bias induced by *zijn* 'his' in stereotypically neutral contexts very early on in the reading process. Extending the analysis to female and male stereotype contexts suggests that the male bias does not persist in otherwise gendered contexts.

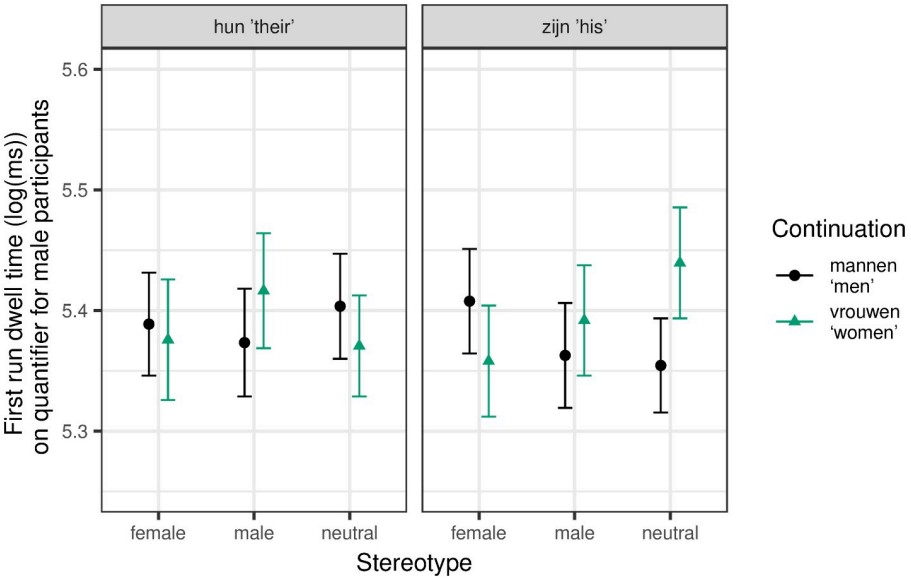

**Fig 2. Mean log-transformed first run dwell time on Region 1 (quantifier) extended to all stereotype contexts for male participants with 95% within-subject confidence intervals based on Morey [60].**

## Discussion

We conducted a conceptual replication of Redl et al.'s [29] eye-tracking experiment, investigating whether the generically-intended masculine pronoun *zijn* 'his' triggers a gender inference and therefore leads to a male bias in language processing. Redl et al. found no evidence for such a male bias. The results of the current experiment, however, do provide such evidence. More specifically, we found an increase in first run dwell time for the earliest region when a female continuation was presented, but for male participants and in stereotypically neutral contexts only. This finding is interesting in several regards.

While we found male participants to exhibit a male bias, no such processing cost occurred for female participants. This suggests that men did not interpret the masculine pronoun generically, while women did. As indicated by a meta-analysis by Henley and Abueg [37], such an asymmetry between men and women has often been found before. For example, Wilson [43] presented participants with statements featuring generically-intended masculine role nouns ending in *-men* (e.g., *salesmen* and *cavemen*). Participants further saw six drawings for each of the role nouns, out of which all drawings that fit the role noun had to be chosen. Three out of the six drawings matched the generic reading of the role nouns, as they either showed a man and a woman, two women or two men, while only the latter drawing matched the non-generic male-specific reading of the role nouns. Wilson found that the male-male drawing was selected in 96% of the cases. The female-male and female-female drawings were only chosen in 75% and 71% of the cases, respectively, indicating that the generically-intended masculine role nouns were more likely to be interpreted as referring to men only. However, Wilson found that female participants selected the female-male and female-female drawings significantly more often than male participants. Other researchers have found similar patterns [7, 8, 11]. Whenever a difference between the two genders was found, it was always the men who showed a biasing effect and the women who interpreted the masculine words as they were intended more often. Henley and Abueg [37] suggest that this asymmetry is rooted in differential language acquisition. This is conceptually similar to postulating a frequency difference between the generic and the male-specific reading for men and women; women have to access the generic reading more often in order to be included and therefore they do so with more ease. This is because women, or rather girls, necessarily have to learn that masculine forms can be used to refer to them too, despite the grammatical gender not matching their own gender. Otherwise they would not be included whenever a generically-intended masculine word is used in reference to them. Boys, on the other hand, will be included as a referent irrespective of whether they interpret the masculine word as male-specific or generic. Therefore, women are more likely to interpret generically-used masculine words the way they are intended, because they have accessed this meaning more frequently in order to be included. Conversely, boys or men will have had to access the generic reading of masculine words less often and this reading will therefore not be as readily available to them. This idea could explain why men showed a processing disadvantage in our experiment while women did not, suggesting that they processed the generically-intended masculine pronoun as it was intended, namely as referring to all genders.

Let us now examine the interplay of pronoun gender and stereotypical gender information in our experiment. Much research has been devoted to the question of how and when grammatical cues, semantic cues and other contextual cues are used in anaphor resolution. Translated to our eye-tracking study, the question thus is whether the pronoun's gender interferes with the processing of the subsequent noun phrase *vrouwen/mannen* 'women/men', and whether gender stereotype information regarding the predicate also plays a role—and at what stage in processing. For example, Esaulova, Reali and Von Stockhausen [61] conducted two

experiments on German role nouns, which were varied between female, male and neutral stereotypes. The role nouns served as antecedents for a pronominal anaphor (*er/sie* 'he/she', Experiment 1) or for the anaphoric expression *diese Männer/diese Frauen* 'these men/these women' (Experiment 2). Esaulova et al. found that the grammatical gender of the role nouns affected early reading times, while an effect of stereotype information was visible in reading times indicative of later processing. They explained these findings in the light of the two-stage reference resolution model by Garrod and Terras [62], which states that anaphors are first linked based on lexical information, with other contextual information only being taken into account later on. Irmen [3] also found that the grammatical gender and the stereotype of the role noun as an antecedent affected reading times at different stages. However, in her experiment, the effect of stereotype actually appeared earlier: stereotype information showed an effect before and after the anaphor, while the grammatical gender of the antecedent had an effect on the anaphor itself. Conversely, Gygax et al. [2] found the masculine grammatical gender of German and French role nouns to override any stereotype information that was additionally provided. In other words, they observed a male bias across all stereotype contexts, the strength of which was not affected by the additional gender stereotype information. Thus, while the evidence is mixed, it is clear that different types of gender information can affect the processing of an anaphor early on.

Similarly, in our experiment, we found evidence for a male bias due to pronoun gender early on, namely for first run dwell time on the pre-view region. We found this effect in stereotypically neutral contexts, but it did not extend to stereotypically male and female contexts. In fact, no difference was found between the female and male continuations in the stereotypically male and female contexts at this stage in processing. Is it possible that both the pronoun's gender and gender stereotypes affected processing simultaneously and led to these results? The result pattern suggests that this is unlikely. If both the gender stereotype context and the pronoun affected the processing of the continuation simultaneously, we would expect the following patterns: For male stereotype contexts, we would expect an even larger increase in reading time for female continuations due to a "double" male bias. since both the stereotype and the pronoun favor a male referent. In stereotypically female contexts, on the other hand, the masculine gender of the pronoun and the female stereotype information could even balance each other out. This is not the pattern we see in our experiment. Instead, we only found a difference in reading time in stereotypically neutral contexts, and no differences in gender-stereotyped contexts; the pronoun thus had an effect in stereotypically neutral contexts, but no other effects—neither of pronoun gender nor of stereotypical gender—were observed in stereotypically female and male contexts at this point in processing. This result pattern can be explained by assuming that the female and male stereotype contexts overrode the grammatical information of the pronoun when processing the first sentence clause featuring both gender cues. Thus, in the case of female and male gender stereotypes, the continuation in the second clause would then be compared against a mental model based only on stereotypical, but no grammatical gender information. This idea is supported by the results reported by Gabriel and Gygax [4]. Their sentence evaluation experiment tested stereotypically male, female and neutral role nouns in Norwegian. Norwegian is very similar to Dutch regarding the grammatical gender system, as both fall somewhere in between that of grammatical gender languages such as German and natural gender languages such as English [4, 28]. Gabriel and Gygax found that the grammatical gender information of Norwegian role nouns led to a male bias, but only when the role nouns were stereotypically neutral. In the case of stereotypically male or female role nouns, the results were guided by the gender stereotype and *not* by grammatical gender. Female and male stereotype information thus overrode the grammatical gender information of the generically-intended role nouns. The authors note that there was a slight indication that

the grammatical gender of the role nouns may have attenuated female gender stereotypes, but the authors warn to be cautious regarding this interpretation due to a small effect size. However, when looking at stereotypically *male* contexts, there was absolutely no additional effect of grammatical gender; the authors conclude that the results were guided by the male gender stereotype alone.

Studies on grammatical gender languages such as French and German suggest that the grammatical gender information of generically-intended masculine role nouns can fully override stereotype information [2, 31]. Thus, like Gabriel and Gygax [4], these authors did not find a "double" male bias when masculine role nouns were stereotypically male. In German and French, referent gender was inferred based on grammatical gender alone, as gender stereotypes were overridden by grammatical gender [2, 31]. In languages such as Norwegian and Dutch, however, the opposite seems to be possible: gender stereotype information seems to be able to override grammatical gender information of (at least some types of) generically-intended masculine words [4]. Thus, the masculine grammatical gender of generically-intended words has previously been found to override gender stereotypes in some languages, but the opposite has been reported for other languages.

If the male and female stereotype contexts indeed overrode the pronoun's grammatical gender in our experiment, we would of course expect to see an effect of the stereotype itself when the gendered noun phrase is encountered. More specifically, we would expect an increase in reading time for noun phrases of which the gender does not match the gender stereotype (i.e., *women* + male stereotype; *men* + female stereotype), or at least for male referents in stereotypically female contexts, as Redl et al. [29] had found. At this early point in processing, we do not observe this. However, Redl et al. only found an effect of a stereotype violation later on in processing; male protagonists engaging in stereotypically female activities led to an increase in regression path duration and dwell time on the gendered noun (i.e., a proper noun in their case). In our current experiment, we solely focused on the effect of the masculine possessive pronoun and its potential interaction with stereotype information. Therefore, we did not analyze the female and male gender stereotype contexts when no effect of the pronoun was found. However, a visual inspection of the later regions and reading times suggests that our data, too, show this asymmetrical stereotype violation effect later on, namely for dwell time on the gendered noun (see S1 Fig). Just like previous studies featuring stereotypical activities [29, 63, 64], only male protagonists engaging in stereotypically female activities led to a mismatch effect, but not the other way around. This asymmetry can be explained by the fact that men are generally penalized more by society for transgressing gender roles [65–68]. Importantly, this does not suggest that only female gender stereotypes trigger gender inferences [see also 29, 63, 64]. Thus, even if the mismatch effect of gender stereotypes is asymmetrical in our experiment, this does not refute the explanation that female and male gender stereotypes overrode the grammatical gender information of the pronoun *zijn* 'his'.

To sum up, the generically-intended pronoun *zijn* 'his' was found to cause a male bias in stereotypically neutral stereotype contexts early on in processing. Since we did not find an effect of the masculine pronoun in female and male stereotype contexts, it appears likely that the stereotype information overrode the pronoun's grammatical gender information. This idea is supported by a highly similar effect caused by stereotypical Norwegian role nouns [4]. The stereotype mismatch effect surfaced later in processing than the male bias of the pronoun in stereotypically neutral contexts, paralleling the time course reported by Esaulova et al. [61].

We conceptually replicated the experiment conducted by Redl et al. [29], who had found no evidence that generically used *zijn* 'his' triggered a gender inference and led to a male bias. In our current experiment, we did find evidence of a male bias, but exclusively for male participants, and only when the context did not provide additional gender stereotype. Thus, while

men experienced a male bias, women appear to have interpreted the masculine pronoun as it was intended, namely as referring to all genders. As discussed in the Introduction, differences between previous studies regarding the strength of the male bias of generically-intended masculine role nouns could suggest that a masculine word is more likely to cause a male bias when it has to be processed more consciously due to the nature of the task. It is therefore possible that a male bias would not only surface for male participants, but also for female participants and across all gender stereotype contexts when employing a method such as the sentence evaluation task. We explored this hypothesis in the second experiment.

## Experiment 2: Sentence evaluation

In this second experiment, we address the question whether a male bias of the Dutch generic pronoun *zijn* 'his' would surface when using a method such as sentence evaluation, which requires participants to process and evaluate the sentences—and thus the masculine possessive pronoun—in a more conscious manner than is the case during mere reading. The sentence evaluation paradigm employed in this experiment is similar to the acceptability judgment task, which has long been a subject of debate among linguists [e.g., 69]. More specifically, there has been a discussion regarding the relationship between the processing and the acceptability of a linguistic stimulus. How representative are explicit stimulus judgments of processing, and which additional factors might come into play when a participant is asked to not only read, but also to judge a sentence? For example, Sprouse [70] found syntactic anomalies to affect acceptability judgments more strongly than semantic anomalies or other processing difficulties. It is thus clear that different methodologies tapping into language processing more or less directly can yield differential results. As outlined in the Introduction, research on the English generically-intended masculine pronouns *he* and *his* has consistently shown that their use leads to a male bias, for example as attested by participants' description of more male mental imagery [e.g., 8] or as attested by a higher number of male protagonists in stories written based on a prompt featuring a generically-intended masculine pronoun [e.g., 7]. Likewise, Gygax et al. [2] conducted sentence evaluation experiments and found that German and French masculine role nouns lead to a male bias across stereotype contexts. Irmen [3], on the other hand, tested German role nouns using eye-tracking and found grammatical and stereotypical gender information to interact, as opposed to grammatical gender overriding gender stereotypes. This could suggest that methods requiring more conscious processing are more likely to yield evidence for a male bias. The results from Experiment 1 suggest that—on average—female language users are capable of processing the masculine possessive pronoun *zijn* 'his' as gender-neutral in the contexts we presented; for male language users, this seems to be possible in stereotypically female and male contexts, where gender stereotype seems to have overridden the male bias of the pronoun. However, it might be that the pronoun's gender leads to a male bias across stereotype conditions and for both genders when performing a different task. In the case of sentence evaluation, it might be that a mismatch between the pronoun's gender and the natural gender of the referents weighs more heavily when participants are asked to evaluate how well the two sentences go together. This could explain the difference in results between Gygax et al. and Irmen, as well as between Experiment 1 reported in this paper and previous research on English generically-intended masculine pronouns, which has usually found both men and women to experience a male bias (even if this male bias was sometimes stronger for men). If this hypothesis is correct, it would be possible for a male bias of the pronoun to emerge not only for male, but also for female participants when performing the sentence evaluation task. More specifically, we would then expect sentences featuring reference to women to be evaluated as bad continuations more often. Furthermore, response times should be

higher when a female referent is featured due to the mismatch with pronoun gender, even if the sentence is evaluated as good. To see if there was some truth to this hypothesis, we conducted a sentence evaluation experiment similar to that of Gygax et al. with the same stimuli that we used in the eye-tracking experiment reported in Experiment 1.

Gygax et al. [2] used the sentence evaluation paradigm in English, German and French to test the effect of grammatical and stereotypical gender information on participants' mental representation of gender. They tested stimuli such as the following:

5. a. *The social workers were walking through the station.*

b. *Since sunny weather was forecast several of the women weren't wearing a coat.*

Participants had to answer whether the sentence in (5b) was a good or sensible continuation to the sentence in (5a). They analyzed the type of response (i.e., yes or no) and the response time of yes-responses. In accordance with their hypothesis, they found that English-speaking participants were influenced by the stereotypical gender of the role noun (e.g., female for *social worker*). Thus, continuations mentioning men were deemed less good or sensible continuations after a stereotypically female role noun, while continuations mentioning women were deemed less good after stereotypically male role nouns. This showed in the type of response participants gave, but not in the response times. In French and German, role nouns are marked for grammatical gender and the masculine form is used as a generic in sentences as in (5a). The results for French and German showed that participants were guided by the grammatical gender of the role noun in their responses, and not by the gender stereotype. Thus, second clauses mentioning men were deemed good or sensible continuations more often across all three stereotype contexts. Response times were significantly shorter for male continuations in German, while this effect did not robustly show in French.

The same authors conducted a follow-up study in which they slightly adapted the stimuli. Garnham et al. [31] inserted a sentence between (5a) and (5b), which gave additional information about the group of people using a 3rd person plural pronoun, e.g.:

6. *They went away.*

In English, the 3rd person plural pronoun *they* does not carry any gender information. In French, however, the appropriate pronoun is *ils*, which carries masculine gender and is another example of a generically-intended masculine pronoun. In German, the available 3rd person plural pronoun is *sie*, the surface form of which is identical to the feminine 3rd person singular pronoun. Garnham et al. [31] wanted to test whether a gender-congruent pronoun (i.e., French *ils*) or a gender-incongruent pronoun regarding its surface form (i.e., German *sie*) can increase (in the case of *ils*) or attenuate (in the case of *sie*) the role noun's male bias. The results for English and, more interestingly, French did not differ from the results by Gygax et al. [2]. That is, Garnham et al. did not find that an additional generically-intended masculine pronoun increased the male bias experienced by French participants. For German, however, they found that the pronoun *sie* attenuated the role noun's male bias. These effects were found for the types of responses given and largely also for the response times. Note, however, that the newly collected data were analyzed together with the data collected by Gygax et al., with *experiment* as an additional factor in the design. Thus, the power for finding a main effect of continuation on response times was naturally higher for Garnham et al. as the data by Gygax et al. was included.

Below, we report the results of an experiment which was similar in design to Gygax et al. [2], but used the stimuli from the eye-tracking experiment reported above. We hypothesized that a

male bias of *zijn* 'his' would at least surface for male participants in stereotypically neutral contexts, but possibly extend to stereotypically female and male contexts and also to female participants due to the task requiring more conscious processing of the generically-intended masculine pronoun. We expected this result pattern to show in the responses provided by our participants, as a male bias had robustly surfaced regarding response type in Gygax et al. The effect did not show robustly in their analyzed response times. We therefore deemed it possible that an effect of the pronoun would show in the response type, but possibly not in response times.

## Materials and method

**Materials.** We used the same 72 stimuli as for the eye-tracking experiment, but shortened them for the purpose of the sentence evaluation experiment in order to make the participants' task clearer. More specifically, the connective maar 'but' and everything following it was removed:

7. *Iedereen was zijn schoenen aan het aandoen, waaronder enkele vrouwen/mannen die al bijna klaar waren om de deur uit te gaan.*

'Everyone was putting on his shoes, among whom some women/men who almost were ready to go out.'

These 72 stimuli were distributed over six conditions instead of twelve conditions as was the case with the eye-tracking experiment; we did not include the six control conditions (*Ze waren allemaal hun y aan het x-en*. . . 'They were all x-ing their y. . .'). The results of the control conditions in Experiment 1 did not reveal an overall male bias (which would have made it impossible to attribute any male bias found for the experimental items to the generically-intended pronoun *zijn* 'his'). We therefore did not deem it necessary to include the control conditions in an experiment using the same stimuli. Furthermore, another reason for including the control items in the eye-tracking experiment was to make sure that any differences in reading times are not solely due to lexical differences between the words *vrouwen* 'women' and *mannen* 'men' regardless of our manipulation. This was less of a concern in the sentence evaluation experiment due to the different method. Furthermore, we wanted to increase the number of stimuli per condition, since conducting an experiment outside of the lab could possibly introduce additional noise. We included 72 fillers. Due to the different nature of the task, these were different from the fillers used in the eye-tracking experiment.

Participants had to evaluate two sentence clauses, which were connected by a comma. They were asked to indicate if the second clause is a good continuation to the first clause. In the case of the experimental stimuli, the most common expected answer was *yes*, indicating a good match. Thus, we constructed 36 fillers which asked for a clear no-response. An example is given in (8). These 36 filler items also allowed us to check whether participants complied with the task and read the sentences attentively. The other 36 fillers asked for a yes-response, as illustrated in (9).

8. *Iemand was de planten aan het bewonderen, waaronder een edelsteen die in Zuidoost-Azië gevonden is.*

'Someone was admiring the plants, among which a gemstone that was discovered in South East Asia.'

9. *Iemand was de honden aan het wassen, waaronder een Golden Retriever die aan het blaffen was.*

'Someone was washing the dogs, among which a Golden Retriever that was barking.'

We calculated each participant's mean rejection rate for the 32 incorrect fillers. Participants had to reject at least 75% (i.e., 27 out of 36) in order to be considered in the analysis. Four lists were created, as every stereotypical activity was embedded in four different stimulus versions, due to the variation of the continuation (*vrouwen* 'women' versus *mannen* 'men') and the variation between the quantifiers *enkele* 'some' and *een paar* 'a few', the latter variation not being experimentally relevant. Participants were pseudo-randomly assigned to one of the four lists. The experiment started with two fillers, but the remaining experimental and filler items were shown in a new fully randomized order for each participant. The equal distribution of participant genders across lists in the final sample (i.e., after removing participants who rejected less than 75% incorrect fillers) was achieved by checking participants' responses to filler items throughout the testing process, whilst not inspecting any of the experimental items, and adjusting the list count in Qualtrics accordingly.

**Participants.**   Ninety-four native speakers of Dutch (49 male) completed the online experiment (age 18–30, M = 20.2). All participants were either students (N = 92) or had already obtained a university degree. None of the participants reported to have dyslexia or other reading problems or had participated in the eye-tracking experiment above. One participant had to be excluded as they correctly guessed that the experiment's purpose was to test whether masculine zijn 'his' leads to a male bias. We calculated the mean rejection rate of the incorrect fillers per participant. Thirteen participants were excluded, as they rejected less than 75% of the incorrect filler items. This left us with the data of 80 participants (40 male, age 18–29, M = 20.1)–with 10 female and 10 male participants for every list.

**Procedure.**   We implemented the experiment in Qualtrics [48]. Participants first received information about the experimental procedure itself as well as general information about the university's policy regarding data storage and participant rights, after which they provided consent. They then answered demographic questions before proceeding to the main part of the experiment. They received more detailed instructions regarding their task and saw two practice items. Similar to Gygax et al. [2], every trial started with the prompt **KLAAR?** *Druk op de spatiebalk. 'Ready? Press the space bar'. After pressing the space bar, participants proceeded to the first clause, ending in a comma. By pressing the space bar again, they proceeded to the second* clause. Participants had to press the space bar again after reading the second clause. They then saw a screen asking **GOED VERVOLG?** Nee (C) *Ja (M)* 'Good continuation? No (C) Yes (M)'. Participants could then indicate their choice by pressing either (C) for no or (M) for yes. This functionality was implemented in Qualtrics through JavaScript. Participants were asked to keep their thumbs on the space bar and their left and right index fingers on (C) and (M) respectively throughout the experiment. After the main part of the experiment, participants were asked to guess the purpose of the experiment. The whole procedure took approximately 30 minutes.

## Analysis

We excluded responses to sentence clauses which had been clicked away within less than 300ms (either sentence clause 1 or sentence clause 2), as we assumed that participants did not properly read these. This led to the exclusion of 3.3% of the data. We then visually inspected a histogram plotting the log-transformed response times in order to identify outliers. Based on this distribution, we decided to remove datapoints with a response time larger than 15000ms (or 15 seconds). This led to the exclusion of a further 13 datapoints.

For the response data, we fitted a mixed effects logistic regression model from the binomial family using the *glmer* function from the *lme4* package [55]. The dependent variable was whether or not participants thought that the second part of the sentence was a good

continuation (coded as 1) or a bad continuation (coded as 0). The fixed effects CONTINUATION, STEREOTYPE and PARTICIPANT GENDER were coded the same as for the eye-tracking experiment. The main difference in design compared to Experiment 1 was that the factor PRONOUN did not apply, since all stimuli featured *zijn* 'his'; this simpler design allowed us to analyze the full data-set (i.e., including stereotypically female and male contexts) from the start. The full random structure permitted by the design was initially included, thus random intercepts for participants and items as well as all permissible random slopes were fitted. Model simplification was done the same way as for the eye-tracking experiment.

We further fitted a linear mixed effects model with the response time of yes-responses as the dependent variable. We defined the response time as the timespan from the moment the second sentence clause became visible until the moment either (C) or (M) was pressed. Following Gygax et al. [2], all no-responses (10.1% of the data after applying the above exclusion criteria) were discarded for this analysis. The response times were log-transformed to render the data more normal. Fitted fixed effects were the same as for the generalized linear model, and so was the approach regarding the random structure. The models for responses as well as response time can be found in the (S2 Model summaries Experiment 2).

## Results

**Response.** The final and best model included random slopes for CONTINUATION for both participants and items, as well as random slopes for STEREOTYPE (female versus neutral) for participants. Descriptively, sentences with male continuations resulted in more yes-responses than sentences with female continuations (see Fig 3), but the main effect of CONTINUATION was far from significant ($\beta = -0.02$, SE = 0.19, z = -0.1, p = 0.92). None of the other fixed effects were significant either.

**Response time.** The results for the response times are shown in Fig 4. The final and best model included random slopes for CONTINUATION for items, as well as random slopes for STEREO-TYPE (male versus neutral) for participants. The model yielded no significant effects.

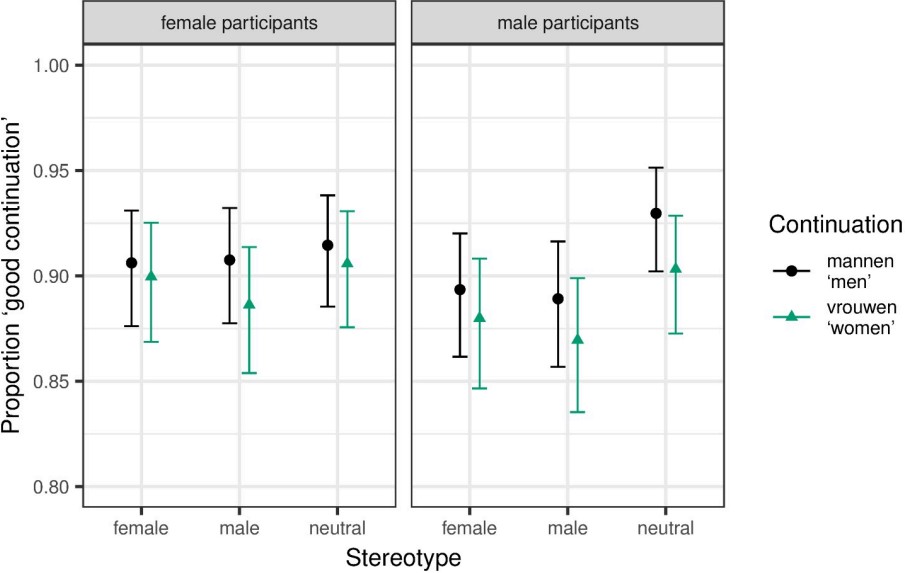

**Fig 3. Mean proportion of sentences of which the continuation was deemed good by participants with 95% exact confidence intervals calculated with the *exactci* function from the *PropCIs* package in R [71].**

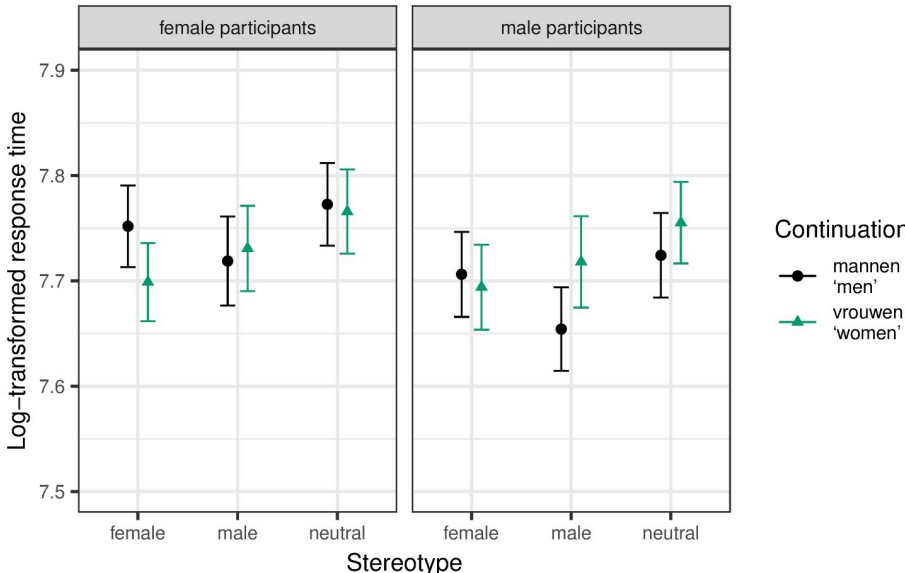

**Fig 4. Log-transformed mean reaction times with 95% within-subject confidence intervals based on Morey [60].**

## Discussion

We conducted a sentence evaluation experiment with stimuli featuring the generically-intended masculine pronoun *zijn* 'his' followed by a reference to either women or men. Participants had to judge whether the clause featuring this reference was a good continuation to the preceding clause featuring *zijn* 'his'. The rationale was that if the generically-intended masculine pronoun was not interpreted as intended and caused a male bias, this would be reflected in a lower acceptance rate of female continuations, as well as higher response times to female continuations even when they were deemed good continuations. However, we did not find that female continuations led to sentences being evaluated as good less often. The hypothesized male bias was also not reflected in response times. This is in stark contrast to the eye-tracking results of Experiment 1, which featured the very same stimuli and provided evidence of a male bias for male participants in stereotypically neutral contexts. We had hypothesized to at least replicate this pattern or even see the male bias extend to other conditions, since the sentence evaluation task requires a different and more conscious processing of the generically-intended masculine pronoun than reading during eye-tracking. However, all continuations scored very high with no significant differences between them. We will discuss possible reasons for these differential results between Experiment 1 and 2 in the General discussion.

## General discussion and conclusion

We conducted two experiments to test whether the generically-intended masculine pronoun *zijn* 'his' causes a male bias. Redl et al. [29] previously conducted an eye-tracking experiment addressing the same question, but did not find evidence for a male bias. The first of the two experiments presented in this paper is a conceptual replication of the Redl et al. study. After having made several improvements to the initial eye-tracking experiment, we did find evidence for a male bias early in processing, but for male participants and in stereotypically neutral contexts only. In a second experiment, we used the sentence evaluation paradigm with the same stimuli to see whether a method asking for explicit judgments would reveal a male bias,

possibly even across stereotype contexts and for male as well as female participants. However, this was not the case. No evidence of a male bias was found in the sentence evaluation experiment. Taken together our results suggest that the generically-intended masculine pronoun *zijn* 'his' can cause a male bias under certain conditions (namely for men and in stereotypically neutral contexts), but that this male bias only surfaces with a highly time-sensitive method tapping into language processing directly, as well as when looking at the earliest processing stages. Conversely, this suggests that the generic reading of the generically-intended masculine pronoun *zijn* 'his' may be readily available for women, at least in contexts similar to the ones we tested; for men, the generic reading appears to be available as well when the pronoun is embedded in stereotypically female or male contexts, as well as during a task requiring participants to go beyond processing and explicitly evaluate sentences featuring *zijn* 'his'.

Based on our results, it thus seems likely that our initial assumption regarding the testing method is incorrect: we did not find an experimental method requiring more conscious processing and evaluation of a generically-intended masculine pronoun to lead to a larger male bias, at least not when comparing eye-tracking and sentence evaluation. However, it is possible that the generic use of *zijn* 'his' would lead to a male bias with the offline methods used by researchers to test English *his* and *he* in the 1970s, 80s and 90s, but not with the sentence evaluation task. As outlined in the Introduction, common tasks in these early experiments were story-writing based on a prompt featuring a generically-intended masculine pronoun or noun [7, 10, 11], while other researchers asked participants to describe their mental imagery after reading or listening to such a prompt [8, 9]. It could be that the pronoun's gender is more likely to have an effect when engaging in such (production) tasks. For example, when being asked to write a story based on a prompt featuring a generically-intended masculine pronoun such as *his* (e.g., *In a large coeducational institution the average student will feel isolated in **his** introductory courses*, as used by Moulton et al. [7]), participants were shown to be more likely to write a story about a man than when the pronoun *their* was featured in the same sentence. This is clearly a sign of a male bias of *his*, but these results do not exclude the possibility that participants would deem stories featuring a woman an equally good example of the prompt in a rating study. It could be interesting to see in future research whether a generically-intended masculine pronoun such as *zijn* 'his' leads to a stronger male bias in such production tasks. Using eye-tracking and sentence evaluation, our participants appear to have been guided less or—in the case of sentence evaluation—not at all by the pronoun's gender compared to early research on English. Based on our results, it rather seems like a sensitive method directly tapping into processing such as eye-tracking is needed in order for the male bias of the generically-intended masculine pronoun *zijn* 'his' to show.

So why exactly does the male bias—against our hypotheses—only surface when using a time-sensitive method tapping into the earliest stages of processing as in Experiment 1, but not in Experiment 2? This could be due to the arguably smaller biasing effect of the pronoun *zijn* 'his' compared to other generically-intended masculine words that have previously been studied. While Experiment 1 provides evidence of a male bias, it does so only for men and only for stereotypically neutral contexts—as gender stereotype information seems to have overridden pronoun gender in stereotypically male and female contexts. Furthermore, the pronoun's male bias only surfaced in the very earliest stages of processing. Particularly this latter finding suggests that participants recuperate from these processing difficulties and adjust the mental gender representation of the referents relatively fast when confronted with information contradicting the male bias. Thus, when given additional time to process a reference to women as is the case in the sentence evaluation experiment, participants might not be affected by this initial male gender representation anymore and respond in a way that is in accordance with the generic reading of the masculine pronoun.

This relatively weak male bias of *zijn* 'his' at which the present experiments hint is in contrast with most of the experimental literature on role nouns, including sentence evaluation experiments on role nouns. As described in the Introduction, generically-intended masculine role nouns have consistently been found to cause a male bias in languages such as French and German for men as well as women and often across stereotype contexts. This was shown with methods as varied as EEG [5], eye-tracking [3], self-paced reading [6], sentence evaluation [2, 31] and also with Moulton et al.'s [7] story writing method [72]. As described by Redl et al., it is possible that role nouns simply cause a larger male bias than the masculine possessive pronoun *zijn* 'his'. At this point, it is unclear whether this is because generically-intended masculine pronouns (or at least the possessive pronoun) are generally more easily interpreted as generic than role nouns, or whether this is a language-specific difference between Dutch, on the one hand, and grammatical gender languages such as French and German, on the other hand. Regarding the latter option, it is clear that feminine and masculine grammatical gender categories are less prevalent in Dutch as spoken in the Netherlands, since they have collapsed into one common gender category for nouns. Due to a development from syntactic to semantic agreement in the domain of pronouns in Dutch, grammatically masculine pronouns enjoy a default status beyond reference to people [15]. As noted in the Introduction, a word such as *bicycle*, which has common gender in Dutch (*de fiets*) is referred to with a masculine pronoun, e.g. *Hij staat in de schuur* (lit. 'He is standing in the shed'). This increased default status of masculine pronouns might also make the generic reading more readily available when the pronoun is used in reference to people [29]. Research on role nouns in Norwegian [4], a language that like Dutch falls in between grammatical gender languages and natural gender languages [28], supports the idea that the male bias of generically-intended masculine words is not equally strong across languages. Extending the research to other generically-intended masculine pronouns in Dutch as well as other languages such as German and French could answer whether *zijn* 'his' is the exception or the rule among generically-intended masculine pronouns.

It is important to note at this point that it was a conscious choice to focus on the possessive pronoun *zijn* 'his'–as opposed to, for example, the personal pronoun *hij* 'he'. We assumed that if we found *zijn* 'his' to cause a male bias, particularly in the contexts we used, then such a male bias was very likely to be found for a more salient pronoun such as *hij* 'he' as well. This is due to the assumed lower salience of *zijn* 'his' as already discussed by Redl et al. [29] and discussed in more detail by Redl et al. [73]. In short, Redl et al. [73] note that a personal pronoun such as *hij* 'he' and also the role nouns as they were used in the experiments described above [e.g., 2, 3, 5, 31] always function as the subject of a clause. The possessive pronoun *zijn* 'his', however, functions as a determiner and is dependent on the head of the noun phrase (e.g., *zijn **veters*** 'his **shoelaces***'). Due to this dependency, the possessive pronoun has been shown to be a less salient cue of the referent's gender [74]. The male bias is therefore less likely to arise based on a masculine possessive pronoun. The fact that we did find such a male bias for *zijn* 'his', albeit only for male participants in stereotypically neutral contexts and when using eye-tracking, suggests that more salient generically-intended masculine words such as *hij* 'he' would also lead to a male bias, possibly even a larger one. This is indeed what the results by Redl et al. [73] suggest. They found that generically-intended *hij* 'he' led to a male bias for male as well as female participants in similar contexts such as the ones used in this experiment. In addition, lower salience could also partly explain this smaller male bias effect of *zijn* 'his' compared to research on role nouns in German and French [e.g., 2, 3, 5, 31].

In a similar vein, the antecedent of *zijn* 'his' could have further increased the masculine pronoun's generic potential. In our stimuli, the referents were introduced by *iedereen* 'everyone', which also functioned as the subject of the sentence. The pronoun *zijn* 'his' was then anaphorically used to link back to the antecedent. The fact that the group was initially introduced as

*everyone*, a grammatically gender-unmarked and inclusive pronoun, might have attenuated the male bias of *zijn* 'his'. It would be interesting to test this in future research. One could, for example, test the male bias of *zijn* 'his' when it is used cataphorically, for example:

10. *Na zijn veters gestrikt te hebben, was iedereen klaar voor het vertrek. Een paar van de vrouwen/men waren zo goed als klaar.*

'After having tied his shoelaces, everyone was ready leave. Some of the women/men were almost ready.'

Both *iedereen* 'everyone' as well as the female and male continuations could then serve as regions of interest, possibly exposing a male bias of *zijn* 'his' that is larger than the one found in our experiment. However, the pronoun *everyone* clearly cannot fully attenuate a gender bias; the male participants experienced it in our eye-tracking experiment after all. Furthermore, gender stereotypes also affected processing, even when the referents were introduced by inclusive *everyone* and the gender stereotype information was only provided later [29]. Moreover, it has been previously suggested that supposedly gender-neutral words such as *people* or gender-unmarked role nouns can lead to a male bias, too [3, 42, 72, 75]. Thus, even if *everyone* is grammatically inclusive as it can be used to refer to all genders, it might still carry a male bias due to androcentrism in society [76].

Crucially, generically-intended masculine pronouns often have male-biased antecedents in the form of role nouns, as in the example from the newspaper we provided at the very beginning. A male bias could also be more likely to arise in such contexts (e.g., *Een student moet zijn scriptie op tijd indienen* 'A student (MASC.) has to hand in his thesis on time'). To sum up, the stimuli we used likely made it harder for a male bias to arise due to the use of the possessive pronoun *zijn* 'his' as well as the indefinite pronoun *iedereen* 'everyone' as its antecedent. Thus, the small male bias we found in the current experiment suggests that the male bias of generically-used pronouns may in fact be stronger in other, less inclusive contexts.

Finally, the fact that the male bias of *zijn* 'his' only surfaced with the more time-sensitive method tapping into online processing also means that the conclusion that women are able to readily access the pronoun's generic reading has to be treated with caution. It is possible that women experience a male bias, too—albeit substantially smaller than that of men. Alternatively, women might be able to overcome the male much faster than men. Our method might simply not be sensitive enough to detect it in that case. Follow-up experiments will have to pay close attention to differences in men's and women's processing of generically-intended pronouns such as *zijn* 'his'.

To conclude, we have shown that generically-intended *zijn* 'his' can cause a male bias. We found this effect for men and when no female or male gender stereotype information was provided. The male bias surfaced in eye-tracking, but not in sentence evaluation, further suggesting that while *zijn* 'his' can lead to a male bias, the generic reading can be accessed under certain conditions, too.

## Supporting information

**S1 Table. Results of rating study for stereotypical activities.**
(PDF)

**S2 Table. Activities used for experimental stimuli with their mean rating and standard deviation on a 7-point scale, with 1 representing female and 7 representing male.**
(PDF)

**S1 Stimuli. Stimuli for Experiment 1 & 2.**
(PDF)

**S1 Pre-tests. Detailed information on Pre-test 1 and Pre-test 2.**
(PDF)

**S1 Model summaries Experiment 1. Fixed-effect coefficients β, their t-scores, uncorrected p-values and the FDR threshold for the fixed effect in question.**
(PDF)

**S2 Model summaries Experiment 2. Fixed-effect coefficients β, their t-scores and p-values shown for response type and response time.**
(PDF)

**S1 Fig. Mean log-transformed first run dwell time on Region 2 (noun) with 95% within-subject confidence intervals based on Morey (2008).**
(EPS)

**S1 Exploratory analyses. Exploratory analyses on the number of regressions into the first clause in Experiment 1.**
(PDF)

## Acknowledgments

We would like to thank Maria van de Groep and Joske Piepers for their help with the stimuli. We would also like to thank the two reviewers for their useful comments.

## Author Contributions

**Conceptualization:** Theresa Redl, Stefan L. Frank, Peter de Swart, Helen de Hoop.

**Data curation:** Theresa Redl.

**Formal analysis:** Theresa Redl, Stefan L. Frank.

**Investigation:** Theresa Redl.

**Methodology:** Theresa Redl, Stefan L. Frank, Peter de Swart, Helen de Hoop.

**Project administration:** Theresa Redl.

**Resources:** Theresa Redl, Stefan L. Frank, Peter de Swart, Helen de Hoop.

**Software:** Theresa Redl.

**Validation:** Stefan L. Frank, Peter de Swart, Helen de Hoop.

**Visualization:** Theresa Redl.

**Writing – original draft:** Theresa Redl.

**Writing – review & editing:** Theresa Redl, Stefan L. Frank, Peter de Swart, Helen de Hoop.

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
