## [Decision Letter · Decision Letter 0]

12 Oct 2020

PONE-D-20-26281

The male bias of a masculine generic pronoun: Evidence from eye-tracking and sentence evaluation

PLOS ONE

Dear Dr. Redl,

Thank you for submitting your manuscript to PLOS ONE. After careful consideration, we feel that it has merit but does not fully meet PLOS ONE’s publication criteria as it currently stands. Therefore, we invite you to submit a revised version of the manuscript that addresses the points raised during the review process.

Your manuscript has been evaluated by two reviewers, who find your manuscript to be well written and its topic interesting and timely. Between them they also raise several points that will need to be addressed before the manuscript can be reconsidered for publication, however. One specific concern is the introduction, which they suggest should be revised so as to include more information about the Dutch gender system, an improved overview of the literature on context effects on pronoun interpretation, and a proper introduction of the "gender of participant hypothesis" that your study investigates. The reviewers also ask you to improve the description of experimental design and motivate your choice of interest regions. I urge you to pay particular attention to the concern about a potential bias in your experimental materials, due to your stimulus items being introduced by the indefinite pronoun iedereen ‘everyone’, which might have biased participants towards a generic reading of the pronoun 'zijn'. Both reviewers suggest that your discussion be revised, to better explain the unexpected absence of a male bias in the male stereotyped condition and in the light of potential biases in your materials. Please also attend carefully to the reviewers' minor comments.

We look forward to receiving your revised manuscript.

Kind regards,

Claudia Felser, Ph.D

Academic Editor

PLOS ONE

Journal Requirements:

Reviewers' comments:

Reviewer's Responses to Questions

**Comments to the Author**

1. Is the manuscript technically sound, and do the data support the conclusions?

Reviewer #1: Partly

Reviewer #2: Partly

2. Has the statistical analysis been performed appropriately and rigorously? 

Reviewer #1: Yes

Reviewer #2: Yes

3. Have the authors made all data underlying the findings in their manuscript fully available?

Reviewer #1: Yes

Reviewer #2: Yes

4. Is the manuscript presented in an intelligible fashion and written in standard English?

Reviewer #1: Yes

Reviewer #2: Yes

5. Review Comments to the Author

Reviewer #1: Review for PLOS ONE

Title: “The male bias of a masculine generic pronoun: Evidence from eye-tracking and sentence evaluation”

Summary:

The authors were interested in examining the gender representations generated by the Dutch masculine pronoun “zijn”. In their first experiment, they used eye-tracking, and in the second experiment, they used a continuation judgement task. Although nothing really came out of the second experiment, some interesting results came out of the first one. Namely, the masculine pronoun (that can be interpreted generically) showed a male bias, but only in early measure (before the mention of men/women appear), in the neutral stereotype condition, and for male participants.

Evaluation:

I find this topic exciting and highly relevant in the current discussions on the effect of language and gender representation. Yet, I have some concerns about the current version of the paper. At times, it could well be the case that I misunderstood something, and if this is the case, I really do apologise, and I would be happy that the authors simply clarify where there is room for (my) misunderstanding. Also, I have the feeling that some of my issues also apply to Redl et al. (2018), which was published. Therefore, some of my concerns may feel unfair, as no reviewer might have. raised them for Redl et al.. Again, it could be that I misunderstood something (in Redl et al., too, to be honest).

Now, if my concerns are not due to some misunderstanding, the question is what to do. For my main concerns, some additional data could be collected, but to be honest, it would be a different experiment. As such, the authors may be able to extensively discuss the issue(s), and set the grounds for additional experiments on “zijn”. I honestly think that this is possible.

Well, let me come now to my main concerns, and then list smaller ones (yet that also need to be addressed, I reckon).

Main concerns

1. My main concern is about the materials, i.e., the sentences that are used. Note that I don’t know Dutch at all, so I am relying on the English translations. Now, I am concerned that the first parts of the sentences, despite the pronoun in the masculine form, are actually priming inclusion. Namely, they use “everyone” as the reference of the pronoun. So really, this primes “his” to be generically understood. A little bit like in Gygax et al. (2012) when participants were explicitly asked to consider the masculine form generically (which resulted in more inclusive representations, although still masculine). Past research has been concerned with generic and specific interpretations of the masculine form when NO context is given (except stereotyped ones). As such, it is really not that surprising that people can see the masculine form as inclusive, especially that the first word of the sentence actually states that it IS inclusive.

I am therefore wondering whether in sentences like these, the pronoun would be treated differently:

When tying his shoes, the women seemed stressed.

[Terwijl hij zijn schoenen strikte, leek de vrouw gestrest.]

Or, if this is actually grammatically wrong:

When the musician tied his shoes, she seemed stressed.

[Terwijl de muzikant zijn schoenen strikte, leek zij gestrest.]

or

When a musician prepares his guitar, she is always concentrated.

[Wanneer een muzikant zijn gitaar voorbereid, ze is altijd geconcentreerd.]

In Experiment 2, this is even increased by the inclusion of “Someone” in the filler items. So the filler items are about an individual, whereas the experimental items are about “Everyone”. In fact, the difference between the fillers and the experimental items could explain why there was no effect in Experiment 2.

2. My second concern is about the regions of interest chosen. I did not understand why the pronoun “zijn” was not also a region of interest. I mean, not necessarily for first run (although this could inform us as to the pre-processing), but mainly as regression fixations. I would imagine that if the system comes across a female mention in the second part of the sentence, whereas it was introduced by a masculine pronoun, it would re-check whether it was really a masculine form. It could even want to check the “everyone” mention (see my main issue). To be on the safe side, I would actually take the whole first part (until the coma), as region of interest. To me, this is very important, especially when looking at regressions in the “their” condition (which I would guess are pretty absent). I think I would use number of regression fixations.

3. My third concern is more likely a misunderstanding. I do apologise, but I got lost as to the model comparisons, especially as you, if I understand correctly, compared models with different sentence locations, and different measures. As such, intuitively, I would actually present sets of analyses, but different for each measure (not only locations). To me, each measure illustrate a different cognitive process (and the authors should be clearer as to why they chose those measures, and to which mental processes they refer to). So dwell time, for example, includes a wrap-up process, whereas first path looks at more spontaneous processes. Not quite sure that comparing different models with different measures makes sense.

4. Finally, the fact that no male bias was found for the male stereotyped condition is problematic to me. The fact that the neutral stereotype condition did get it (even only for male participants) makes sense, but in this case, why would the male stereotype condition not have it. This is rather odd. The only difference with the neutral condition is that it increases the sense of maleness. If anything, it should be stronger than the neutral condition. I really do not understand this, and it hints at the fact that something was maybe rather wrong in the data, or the way it was handled. See my previous points maybe. Note that the discussion given by the authors is confusing, as they do not clearly say why they did not find the male bias in the male stereotyped context. At least I did not get it. If one thinks that there are two possible biases, then one of them should lead reading. For the male stereotype context, both male biases should result in the same pattern. But it did not. It’s really puzzling to me.

These are my smaller concerns:

5. It would be nice for readers that do not know Dutch so well to explain in more details the grammatical gender properties in Dutch. Maybe the Language Index paper by Gygax, Elmiger, Zufferey et al., could serve as a base.

6. P3: The authors mention “masculine generics”. Does it mean these are different to masculine specifics? Or are they just masculine forms, used or interpretable as generics? I find the terms “masculine generics” misleading (although I have at times also used it)

7. P3:L9: I’m not sure why it is “clearly”. I don’t see anything in the sentence that would give me such a clear context. At least, to me, it is less clear than if the sentence was introduced by “everyone”, which is the case in the present study.

8. P3:L10: The authors mention that “…has occupied linguists for decades…”. Looking at this list (which appeared recently after some French linguists said that no research was conducted on inclusive language), I would say that linguists are not the only ones to address these questions (not that a lot of the authors mentioned in the paper are actually psychologists, or psycholinguists): https://osf.io/p648a/?view_only=a385a4820769497c93a9812d9ea34419

9. P3:L23: a good/sensible continuation. I would add “sensible”

10. P3:L26: “therefore taps less into online processing”: This is an old debate about online vs offline processes. I think that this debate is not so informative. Even if you measure eye movements. When making judgements about sentences, i.e., what is considered off-line, people use their mental models to answer, and these mental models were also built on-line. Anyway, I would concentrate on early processes versus later processes. But that’s maybe just my opinion.

11. P5:L59: the reference should be [15], right?

12. P5: While reading, I kept asking myself about what measures the authors used for their eye-tracking. Even when they present Redl et al.’s study, I was wondering what measures they took: dwell time, first fixation, number of fixation, spill over regions,… (it is only mentioned on line 123). See my comment later on the choices of measures in the present study.

13. P6: L94: To my knowledge, except from the study they cite here (and maybe a handull of others), none of the other studies did find any gender of participant effect. It might be worth mentioning this.

14. P6: L88: The authors seem to say that Redl et al. needed improvement in the design. It would be good to say early what the problems were.

15. P7: I really like the improvement made in the continuation sentence, as, although I like the Redl et al. paper, I did find that in Redl et al., the continuations may be quite easily dissociated with the first sentences. In fact, this was also criticised in Gygax et al. but the authors did run a pilot study to ensure that this was not the case.

16. P8: L145: I am a little puzzled by the gender of participant hypothesis. Of course, it is based on several studies on the pronoun, but I am wondering what would the underlying mechanism be. It could be that as women are exposed to masculine forms when they are referred to (or when they would like to be referred to), they are more used to consider it as generic. Actually, the authors discuss that later on. It could be introduced early to consolidate their hypothesis.

17. P8: L148: Presenting method before participants is rather odd, right?

18. P9: L161: This is just a comment, but the fact that scale direction affects stereotype rating does not have to be controlled in this case, as differences between the role noun stereotypicality (or between activities) remains constant, irrelevant of scale direction). What changes are the overall scores.

19. P17:L82: I think that Gygax, Lévy et al., 2019, found that actually, the male bias was more pronounced for girls (between 3-5 years old)

20. P22: L519: good/sensible. Adding “sensible” is important, as in earlier work in English (Garnham, Oakhill, Reynolds), the term “sensible” was used, and in Gygax et al., the French translation was something between good and sensible.

21. Depending on the changes made according to the issues raised, I would imagine the discussion to change substantially.

Reviewer #2: The paper presents two experiments that examined whether the Dutch possessive pronoun zijn ‘his’ may be perceived and interpreted as gender specific despite its intended generic use. Two methodologically different approaches employed in the experiments tap into automatic sentence processing during reading (eye-tracking, Experiment 1) and more conscious processing (sentence evaluation, Experiment 2). The results show evidence for a male bias in neutral contexts and in male participants only in Experiment 1, while no such evidence is found in Experiment 2. The authors conclude that a male bias in the interpretation of the masculine generic pronoun zijn ‘his’ can only surface during online processing, while the generic reading is accessed otherwise.

The paper is generally well-written and has a clear argumentation line. I believe, a few points/references could be added to make the introduction more comprehensive and the discussion more complete. While experimental materials were thoroughly constructed, I also have several questions regarding the methodology, as well as some reservations concerning the authors’ conclusion.

I believe that readers would benefit from some more information on the Dutch gender system in the Introduction. The authors mention its peculiarities later on in the Discussion, however, to better understand the research question and the motivation of the study it needs to be explained earlier. I would also like to see a more comprehensive overview of the relevant literature on the effects of context on pronoun interpretation as part of the Introduction (e.g., Duffy & Keir, 2004). Finally, since the participants’ gender is at the core of the main finding, introducing research on differences (or the lack thereof) between men and women in their perception of gender biases would certainly be useful. This could also be used to provide more theoretical grounding for why participants’ gender was included in the analysis in the first place.

A methodological concern related to the previous comment: the authors state that they “strove to balance their sample between female and male participants”. However, the number of female and male participants is not equal. Since participants’ gender is related to the main finding of the study, this should be justified.

Even though the Experiment 1 is described in great detail, I found it difficult to understand its design. More precisely, it is not quite clear to me why the authors talk about six (rather than one) “control” conditions and why it was necessary to treat them as such in the analyses. Furthermore, how can we be certain that “their” does not potentially carry a male-bias as well? My next comment addresses a related concern.

Since Experiment 1 is in fact a replication – with some modifications – of Redl et al. (2018), which yielded different results, I could not help wondering about the power of the study. This is especially relevant given a large number of conditions (6 experimental + 6 control). Brysbaert & Stevens (2018), for instance, recommend that a properly powered experiment with repeated measures has at least 1600 observations per condition. While it is not typical for eye-tracking studies to follow this recommendation, I would like to know what the decision about the number of participants/stimuli was based on.

The authors mention that data were log-transformed. However, they do not mention what motivated this particular transformation. What procedure was used to determine which transformation is most beneficial for their data to render it more normal (e.g. Box-Cox)?

The authors make an important point in the Discussion (lines 716-717) stating that “a sensitive method is needed… for the male bias… to show”. Although they do not elaborate on this any further, it raises a potential issue for the methodological approaches in the study of gender biases in general. In fact, some gender biases have previously been overlooked due to their subtle nature (e.g., Esaulova, Reali, & von Stockhausen, 2017). Importantly, simply because some research tools are not sensitive enough to capture these biases (e.g., sentence evaluation in Experiment 2), does not mean that they are not there. This is why I am not convinced by the authors concluding that “the generic reading is often accessed, too”. This issue should certainly be extended in the discussion.

Minor comments:

- lines 114-115: please make explicit why introducing the connective that indicated membership status was important (to establish a reference?).

- line 139: please explain how exactly control conditions allowed to more clearly identify a possible effect of the masculine pronoun.

- line 179: in the

- line 191: “had a mean rating of 3” - please specify according to which scale

- line 208: explain the role of control items (to provide a baseline?)

- line 228: participants’ number is stated as 121 here but 120 in the abstract, please be consistent

- line 278: please explain why a male bias would be most clearly seen in neutral stereotype contexts (and not in female)

- line 304: add “(FDR)” here after “false discovery rate” before using the abbreviation later on

- lines 343-345: “Notably,…” – please check grammar here

- line 349: not a lead

- lines 497-501: the sentence starting with “Even if…” is too long, please rephrase

- line 594: add “order” after “randomized”

- line 608 & 609: replace “second sentence” with “second clause”

6. PLOS authors have the option to publish the peer review history of their article (what does this mean?). If published, this will include your full peer review and any attached files.

Reviewer #1: No

Reviewer #2: **Yes: **Yulia Esaulova

---

## [Author Response · Author response to Decision Letter 0]

13 Nov 2020

Dear Dr Felser and reviewers,

We would like to thank you for the diligent and elaborate feedback. We feel that this constructive criticism helped us to further improve the paper, and we hope that you will find all points of criticism sufficiently addressed.

Please refer to the updated cover letter to see how we incorporated the feedback. All points which were raised are in gray and italics font and we added our response in regular black font for better readability.

Best regards,

Theresa Redl (also on behalf of Dr De Swart, Dr Frank and Prof. Dr. De Hoop)

---

## [Decision Letter · Decision Letter 1]

11 Dec 2020

PONE-D-20-26281R1

The male bias of a masculine generic pronoun: Evidence from eye-tracking and sentence evaluation

PLOS ONE

Dear Dr. Redl,

Thank you for submitting your manuscript to PLOS ONE. After careful consideration, we feel that it has merit but does not fully meet PLOS ONE’s publication criteria as it currently stands. Therefore, we invite you to submit a revised version of the manuscript that addresses the points raised during the review process.

Both reviewers appreciate the thoroughness of your revisions and agree that your manuscript is now much stronger. However, Reviewer #1 still has some concerns, which I hope you will be able to address in another round of revisions. Their main concern is your use of "everyone" in the context sentences. While you have explained at some length why you did not use role nouns - and I think your reasons for this are perfectly valid - the reviewer remains concerned about the inclusive nature of "everyone". I agree with the reviewer that because of this inclusive reading, your context sentences should not be presented as neutral context sentences, and that you should change the framing of your study accordingly. The reviewer emphasizes that this would by no means invalidate your results or make your study any less interesting, but some rewriting will be neccesary to adequately accommodate the reviewer's point. Please also attend to the reviewer's other points in your revisions.

We look forward to receiving your revised manuscript.

Kind regards,

Claudia Felser, Ph.D

Academic Editor

PLOS ONE

Reviewers' comments:

Reviewer's Responses to Questions

**Comments to the Author**

1. If the authors have adequately addressed your comments raised in a previous round of review and you feel that this manuscript is now acceptable for publication, you may indicate that here to bypass the “Comments to the Author” section, enter your conflict of interest statement in the “Confidential to Editor” section, and submit your "Accept" recommendation.

Reviewer #1: (No Response)

Reviewer #2: All comments have been addressed

2. Is the manuscript technically sound, and do the data support the conclusions?

Reviewer #1: Partly

Reviewer #2: Yes

3. Has the statistical analysis been performed appropriately and rigorously? 

Reviewer #1: I Don't Know

Reviewer #2: Yes

4. Have the authors made all data underlying the findings in their manuscript fully available?

Reviewer #1: No

Reviewer #2: Yes

5. Is the manuscript presented in an intelligible fashion and written in standard English?

Reviewer #1: Yes

Reviewer #2: Yes

6. Review Comments to the Author

Reviewer #1: Summary:

I think the authors did a great job at responding to most comments. And the paper has definitely strengthened. However, there is still, to me, an issue about the use of “Everyone” in the priming sentence. I think that I was not clear in my previous comment about this. I truly apologize for this. It’s not so much that I think the materials are not adequate, or that they should replace "everyone" with role nouns, but I think that the way that some of the arguments and some of the interpretation are framed need to be changed.

I will try, this time, to be clear. In a nutshell, I think that the authors should carefully think about the difference between a “neutral context” (i.e., no gender) and an “inclusive context” (i.e., both gender, or all genders for that matter). My impression is that the authors think that they tested the first context, whereas I would argue that they tested the second. As such, I think that the paper is still very interesting, as it means that when the context is explicitly made inclusive (by using “everyone”), readers can consider the masculine pronoun as generic (actually, not all readers, as men still struggle). Whether the masculine form can still be considered generic in a neutral context (i.e., when absolutely no gender context is present), this is still to be tested.

Let me explain my argument:

The authors say already in the abstract “no other gender information is provided”, but I think that this is somehow wrong. There IS some gender information in the primes, i.e., “everyone”. So the gender information provided is INCLUSIVE, i.e., all genders (and not “no gender”). This is why most gender inclusive language guides in gender marked languages probably include “everyone” as an inclusive form. Maybe the authors misunderstood me the first time, but I am still a little surprised that the authors would not consider “Everyone” as an explicit gender inclusive prime. This is probably why readers show early male bias effect, that may then be overcome by the gender information provided by everyone. As such, the title of the paper should be: “The male bias introduced by the masculine form when primed as generic”. This is very important. In fact, the term “prime” is important. In the same way, on page 7, line 120, when presenting Redl et al., the authors should say that the possessive pronoun was “…in a generically-intended priming contexts…”. The fact that the sentences presented “everyone” before “his” is priming inclusiveness (and not neutrality).

When you think about it, it’s very much like the second study of Duffy and Keir, in which the authors present some context that enables readers to go beyond stereotypes. So in the present experiment, the authors present a strong inclusive context (and not a neutral context, meaning no gender) that enables readers to go beyond the maleness activated by the grammatical form.

I also bet that if you were to actually inverse somehow the order “everyone – his” to “his – everyone”, there would be a slow down on everyone (i.e., going from masculine to inclusive), showing that everyone IS a gender cue (i.e., it sets a gender context). My point is that “Everyone” is a biasing gender information. It’s not a problem per se, and this is why in gendered languages “everyone” is at times presented as a good inclusive option when referring to people.

Now, a TRUE absence of “other biasing gender information” would be a sentence like (a):

(a) “Sitting on the stairs tying his shoelaces, Maaike (f)/Stefan (m) was also getting ready…”

Essentially, I have the feeling that the authors mistake the activation of an “inclusive gender context” (or inclusive context) with a “no gender context”. If sentence (a) is not possible in Dutch, this means that the masculine pronoun can only be considered as generic IF (and only if) a strong gender inclusive prime such as “everyone” is presented, but not in a no gender context, because in this case, "his" has to be followed by a masculine name.

Again, it is not a problem as such, but it changes the way the data are presented, analyzed and interpreted (also in Redl et al., 2018). This is true because I do not believe that the results from Redl et al. truly constitute a null results. If a context primes inclusiveness (and not neutrality), then you would expect such a results. Now, if the context had been truly neutral (see example (a), although in Dutch it may not be possible), then it would constitute a null result.

In terms of results, this means that even when the context primes an inclusive representation of the masculine pronoun (and not just a neutral context), male participants still cannot do it. For them, “everyone” is masculine, which to me, is typical of an androcentric perspective. This may mean, and it is related to my issue, that the effects are not necessarily bound to the pronoun, but also to how "everyone" is interpreted.

I hope I am clearer this time.

General comment:

Before giving some rather minor points, I would like to stress another issue that I have also found in the literature on this topic. I think that we should stop using the terms “masculine generics” in this literature. I know it has been used very often (and by our lab too), but I think (now) that this is fundamentally wrong. Essentially, there is no such thing as a “masculine generic pronoun” (or noun). There is a masculine pronoun (or noun), that’s it. Now, it can be interpreted as generic or as specific, but there is no difference between what would be called a “masculine generic pronoun” or a “masculine specific pronoun”. Both are the same lexical item. I think Hamilton (1988) was wrong to name it this way. It’s highly misleading for readers that do not have grammatical gender in their languages. They will actually think that there are two lexical items (like “han” and “hen” in swedish). There’s only one, the “masculine form”. I hope that the authors see what I mean. In all mentions of "masculine generic form", the authors could simply take out the "generic".

Minor comments:

1. P4:L50: I find it odd that the authors refer to Norwegian, as to my understanding at least (and according to Gygax et al., 2019 and others), Norwegian is special, as the masculine form was dropped in the 1980’s. If I am not mistaking, this was not the case in Swedish (which adopted a third – neutral - pronoun)

2. P5:L56-57: The authors mean ‘female writer’, ‘female teacher’…right? Or maybe I missed something.

3. P5:L65: This is interesting, as the male bias is still present in Norwegian (Gabriel et al., 2008) – for non-stereotypical role nouns –, although there is no “feminine” contrast anymore (at least in the main official language)

4. P6:L85: I don’t understand the sentence. Did the author mean “…tested the effect of gender stereotype information and the effect of the masculine form on the representation of gender”?

5. P6:L86: I would just put “…similar to the grammatical gender – the gender stereotypicality…”

6. P6: L91: “…the stereotype OF THE ROLE NOUN, as..” and not of the verb.

7. P6: L93: Remove “actual”. Same on line 103.

8. P7: L112: The references 34-36 are somehow not really related to Spanish, right?

9. P19:L420-422: I would say that this is an empirical question. By saying this, does it mean that the authors did analyse regression path to the first clause? And that the results were the same. Given the rather lack of effects, I think it would be reassuring to know that the results are the same (even by presenting some Supplementary analysis in a different file.

10. P19:L429: I do find it rather odd that the analysis was conducted separately across stereotype groups. Did Redl et al. do the same? And is this why the authors used the Benjamini and Hochberg procedure. This could be a little clearer I guess.

Reviewer #2: (No Response)

7. PLOS authors have the option to publish the peer review history of their article (what does this mean?). If published, this will include your full peer review and any attached files.

Reviewer #1: No

Reviewer #2: **Yes: **Yulia Esaulova

---

## [Author Response · Author response to Decision Letter 1]

11 Mar 2021

The response to the reviewer has been included in the new cover letter.

---

## [Editor Report · Decision Letter 2]

16 Mar 2021

The male bias of a generically-intended masculine pronoun: Evidence from eye-tracking and sentence evaluation

PONE-D-20-26281R2

Dear Dr. Redl,

We’re pleased to inform you that your manuscript has been judged scientifically suitable for publication and will be formally accepted for publication once it meets all outstanding technical requirements.

Kind regards,

Claudia Felser, Ph.D

Academic Editor

PLOS ONE
---

## [Editor Report · Acceptance letter]

23 Mar 2021

PONE-D-20-26281R2 

The male bias of a generically-intended masculine pronoun: Evidence from eye-tracking and sentence evaluation 

Dear Dr. Redl:

I'm pleased to inform you that your manuscript has been deemed suitable for publication in PLOS ONE. Congratulations! Your manuscript is now with our production department. 

Kind regards, 

on behalf of

Dr. Claudia Felser 

Academic Editor

PLOS ONE